# The BRCA2-MEILB2-BRME1 complex governs meiotic recombination and impairs the mitotic BRCA2-RAD51 function in cancer cells

Jingjing Zhang[1,10], Manickam Gurusaran [2,10], Yasuhiro Fujiwara [3,10], Kexin Zhang[1], Meriem Echbarthi[1], Egor Vorontsov[4], Rui Guo[5], Devon F. Pendlebury[6,7], Intekhab Alam[6,9], Gabriel Livera[8], Martini Emmanuelle[8], P. Jeremy Wang [5], Jayakrishnan Nandakumar [6,7], Owen R. Davies [2] & Hiroki Shibuya [1✉]

Breast cancer susceptibility gene II (*BRCA2*) is central in homologous recombination (HR). In meiosis, BRCA2 binds to MEILB2 to localize to DNA double-strand breaks (DSBs). Here, we identify BRCA2 and MEILB2-associating protein 1 (BRME1), which functions as a stabilizer of MEILB2 by binding to an α-helical N-terminus of MEILB2 and preventing MEILB2 self-association. BRCA2 binds to the C-terminus of MEILB2, resulting in the formation of the BRCA2-MEILB2-BRME1 ternary complex. In *Brme1* knockout (*Brme1*$^{-/-}$) mice, the BRCA2-MEILB2 complex is destabilized, leading to defects in DSB repair, homolog synapsis, and crossover formation. Persistent DSBs in *Brme1*$^{-/-}$ reactivate the somatic-like DNA-damage response, which repairs DSBs but cannot complement the crossover formation defects. Further, MEILB2-BRME1 is activated in many human cancers, and somatically expressed MEILB2-BRME1 impairs mitotic HR. Thus, the meiotic BRCA2 complex is central in meiotic HR, and its misregulation is implicated in cancer development.

[1] Department of Chemistry and Molecular Biology, University of Gothenburg, SE-40530 Gothenburg, Sweden. [2] Institute for Cell and Molecular Biosciences, Newcastle University, Newcastle upon Tyne, UK. [3] Institute for Quantitative Biosciences, University of Tokyo, 1-1-1 Yayoi, Tokyo 113-0032, Japan. [4] Proteomics Core Facility, Sahlgrenska Academy, University of Gothenburg, SE-40530 Gothenburg, Sweden. [5] Department of Biomedical Sciences, University of Pennsylvania School of Veterinary Medicine, Philadelphia, PA 19104, USA. [6] Department of Molecular, Cellular, and Developmental Biology, University of Michigan, Ann Arbor, MI 48109, USA. [7] Program in Chemical Biology, University of Michigan, Ann Arbor, MI 48109, USA. [8] Laboratory of Development of the Gonads, UMRE008 Genetic Stability Stem Cells and Radiation, Université de Paris, Université Paris Saclay, CEA, F-92265 Fontenay aux Roses, France. [9] Present address: Department of Physiology and Biophysics, Case Western Reserve University, Cleveland, OH 44106, USA. [10] These authors contributed equally: Jingjing Zhang, Manickam Gurusaran, Yasuhiro Fujiwara. ✉email: hiroki.shibuya@gu.se

Breast cancer susceptibility gene 2 (*BRCA2*) is a tumor suppressor involved in mitotic homologous recombination (HR)[1], and mutations in this gene predispose to a variety of adult and pediatric cancers[2,3]. The primary function of mitotic BRCA2 is thought to be the loading of recombinase RAD51 onto DNA double-strand breaks (DSBs), which promotes their homology-directed repair[4,5].

Unlike mitotic HR that repairs accidental DSBs, the meiotic HR is a highly regulated process and is needed for the repair of programmed DSBs, which subsequently promotes homologous synapsis and crossover formation[6]. Most of what is known about the molecular scheme of meiotic HR has come from studies in unicellular model systems, and mammalian-specific factors have been the target of much effort in an attempt to understand the corresponding mechanisms in higher eukaryotes.

In a recent study in mice, we identified meiotic localizer of BRCA2 (MEILB2/HSF2BP), a germ-cell specific co-factor of BRCA2 that binds to the evolutionarily conserved but functionally uncharacterized MEILB2-binding domain (MBD) of BRCA2[7]. The loss of BRCA2 at DSBs in *Meilb2*[−/−] male mice leads to loss of meiotic recombinase localization and subsequent sterility. Although the identification of MEILB2 shed light on the integral roles of BRCA2-MEILB2 in meiotic HR, how BRCA2 switches its roles from mitotic to meiotic HR and mediates meiosis-specific events, such as homologous synapsis and crossover formation, has been largely unclear.

In order to clarify the meiosis-specific modification of BRCA2, we screen for MEILB2-interacting proteins in murine germ cells and identify BRCA2 and MEILB2-associating protein 1 (BRME1). We find that BRCA2-MEILB2-BRME1 forms a stable ternary complex specific to meiosis, and in vivo genetic analyses clarify the mechanism that governs the assembly of BRCA2-MEILB2-BRME1 on meiotic ssDNA and the essential function of BRCA2-MEILB2-BRME1 in meiotic DSB repair, homologous synapsis, and crossover formation. Further, we demonstrate that *MEILB2-BRME1* is a potential proto-oncogene that impairs mitotic BRCA2 functions, in sharp contrast to its meiotic roles.

## Results

**BRME1 is a meiosis-specific MEILB2-interacting protein**. To identify MEILB2-binding proteins that regulate BRCA2 in meiotic DSB repair, we performed yeast two-hybrid (Y2H) screening of a mouse testis cDNA library. Along with BRCA2, a functionally uncharacterized protein coded by *4930432K21Rik* was most frequently identified (Fig. 1a). The *4930432K21Rik* gene is evolutionarily conserved in vertebrate species (Supplementary Fig. 1a). The expression of *4930432K21Rik* was specifically upregulated in germ-line tissues, similar to the expression of *Meilb2* (Fig. 1b). We named this conserved meiotic gene product BRME1 (BRCA2 and MEILB2-associating protein 1).

Fourteen BRME1 peptides located at the C-terminus of BRME1 were identified as MEILB2-binding peptides (Fig. 1c). We named the common 82 amino-acid region the MEILB2-binding domain (MBD), and Y2H analysis confirmed that BRME1-MBD was necessary and sufficient for MEILB2 interactions (Fig. 1d). The FLAG pulldown assay from cultured cells co-expressing FLAG-tagged BRME1 truncations ([FLAG]-BRME1) with MYC-tagged MEILB2 (MEILB2[-MYC]) also proved that BRME1-MBD is necessary and sufficient for an interaction with MEILB2 (Fig. 1e).

**BRME1 stabilizes MEILB2 by preventing self-association**. We purified the recombinant MEILB2-BRME1-MBD heterocomplex from *Escherichia coli* and found that the α-helical N-terminus of MEILB2 (α1+2) was sufficient for BRME1 binding (Fig. 1f and

Supplementary Fig. 1b). α1 (a.a. 18–55) alone bound to BRME1 while α2 (a.a. 51–122) did not, suggesting that α1 is necessary and sufficient for the BRME1 interaction (Fig. 1f and Supplementary Fig. 1c). Circular dichroism (CD) showed that BRME1 increased the melting temperature of MEILB2 α1+2 from 29°C to 50°C (Fig. 1g). The complex with α1 was even more stable, with a melting temperature of 74°C (Fig. 1g), suggesting that BRME1 binding confers structural stability to MEILB2.

Size-exclusion chromatography multi-angle light scattering (SEC-MALS) showed that the complexes between BRME1 and MEILB2 (both α1 and α1+2) were in a clearly 2:2 stoichiometry (Fig. 1h, i and Supplementary Fig. 1d). However, in the absence of BRME1, both α1 and α1+2 showed a tendency to oligomerize – α1+2 formed either dimers or octamers and α1 formed a tetramer (Fig. 1h, i and Supplementary Fig. 1d)). This suggests that the BRME1-binding site mediates self-association in the absence of BRME1.

The small angle X-ray scattering real-space $P(r)$ interatomic distance distribution of the α1+2 dimer showed a positive skew, indicating an elongated molecule with a maximum dimension of 160 Å, closely matching the theoretical length of this sequence as an elongated α-helix (Fig. 1j and Supplementary Fig. 1e, f). Similarly, the ab initio dummy atom model indicated an elongated molecule with an acute-angle kink towards one end (Fig. 1k), suggesting that the α1+2 dimer adopted an elongated coiled-coil-like structure with a hinge possibly between α1 and α2. In support of this, monomeric α2 had a length and thickness consistent with it forming one helix of the longer arm of the α1+2 dimer (Fig. 1j, k). Notably, the α1+2+BRME1 2:2 complex had a similar $P(r)$ distribution as the α1+2 dimer (Fig. 1j) with a similar kink (Fig. 1k), suggesting that BRME1 binds to an existing coiled-coil-like α1+2 dimer. In contrast, the α1+2 octamer had only a slightly increased length but considerably more bulk than the dimer (Fig. 1j) with thicker arms (Fig. 1k), suggesting that α1+2 forms higher-order assemblies through lateral interactions in the absence of BRME1. Taken together, we conclude that BRME1 binds to an elongated coiled-coil-like MEILB2 α1+2 dimer, stabilizing this structure in a 2:2 complex and blocking its ability to form higher-order assemblies through lateral self-association (Fig. 1l).

**BRCA2, MEILB2, and BRME1 form a ternary complex**. MEILB2 interacts with BRME1 along with BRCA2, implying that they can form a ternary complex. Indeed, the BRCA2-binding region of MEILB2 is in its C-terminus with an armadillo-repeat domain that contains arginine 204, which corresponds to R200 in human, the mutation of which abolishes MEILB2-BRCA2 interactions[8] (Fig. 2a). We purified recombinant MEILB2 C-terminus (87–338 a.a., 27.9 kDa in theory), BRCA2-MBD fragment (2219–2285 a.a., 7.4 kDa in theory), and their heterocomplex. Not only the individual proteins (Supplementary Fig. 2a), but also their stoichiometric heterocomplex were successfully purified from *E. coli*, which showed clear single-peaked elution profiles from size-exclusion chromatography (Fig. 2b). SEC-MALS estimated the molecular weight of the MEILB2 C-terminus and BRCA2-MBD to be 57 kDa and 7 kDa, respectively, suggesting that the MEILB2 C-terminus forms homodimers, whereas BRCA2-MBD presents as monomers (Fig. 2c). Surprisingly, SEC-MALS determined the heterocomplex to be 118 kDa, suggesting that the MEILB2 C-terminus homodimer is converted into a tetramer in the presence of BRCA2 (e.g. the theoretical molecular weight of a 4:2 MEILB2-BRCA2 complex is 126 kDa; Fig. 2c). BRCA2 seems to stabilize MEILB2 in its tetrameric state, because the heterogenous oligomeric population of MEILB2 observed in isolation (multiple peaks in the orange curve in Fig. 2c) disappears in the presence of BRCA2 (black curve in Fig. 2c).

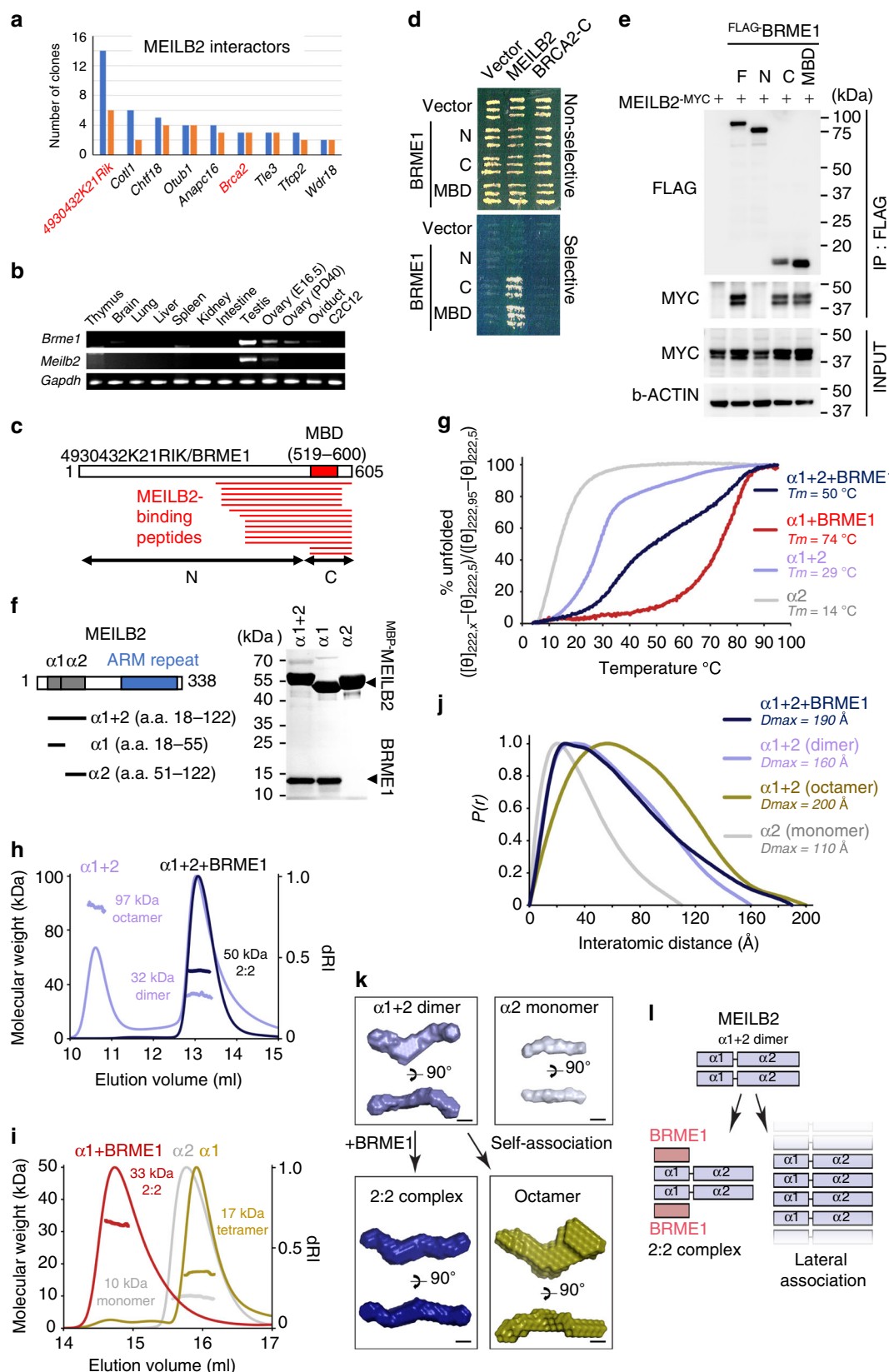

To confirm the existence of a ternary complex, we expressed GFP-tagged BRCA2 ([GFP]-BRCA2) truncations together with [FLAG]-BRME1 and MEILB2[-MYC] and performed a GFP pulldown assay. Consistent with our previous findings[7], [GFP]-BRCA2-C, but not N and M, specifically interacted with MEILB2[-MYC] (Fig. 2d,

lane 9). In contrast, none of the [GFP]-BRCA2 truncations interacted with [FLAG]-BRME1, suggesting that there is no binary interaction between BRCA2 and BRME1 (Fig. 2d, lane 2, 5, 8). However, when all three proteins were co-expressed, we detected the formation of a [GFP]-BRCA2-C-MEILB2[-MYC]_[FLAG]-BRME1

**Fig. 1 Identification of BRME1. a** Genes identified in the MEILB2 Y2H screening. Blue and red bars indicate the number of total and original clones, respectively. Genes in red indicate genes involved in this study. **b** Tissue-specific expressions of *Brme1*, *Meilb2*, and *Gapdh* (loading control) shown by RT-PCR. C2C12 is a myoblast cell line. *E* embryonic day. *PD* postnatal day. **c** Mapping of BRME1 peptides identified in the Y2H screening. MBD (a.a. 519–600) is the common region in all peptides. BRME1-N (a.a. 1–518) and -C (a.a. 519–605). **d** Y2H interactions. BRME1-N (a.a. 1–518), -C (a.a. 519–605), or -MBD (a.a. 519–600) were used as prey. MEILB2 and BRCA2-C (a.a. 2036–3329) were used as bait. **e** IP with the FLAG antibody from B16-F1 cells expressing MEILB2-MYC and FLAG-BRME1 truncations; F (a.a. 1–605), N (a.a. 1–518), C (a.a. 519–605), and MBD (a.a. 519–600). **f** Schematic of the MEILB2 sequence highlighting the recombinant protein constructs with amylose pulldown following co-expression of BRME1-MBD (a.a. 519–600) with MBP-MEILB2 α1 + 2, α1, and α2. **g** CD thermal denaturation, recorded as percent unfolded based on the helical signal at 222 nm, with melting temperatures estimated as shown. **h, i** SEC-MALS analysis. **h** α1 + 2 + BRME1 is a 50 kDa 2:2 complex (theoretical ~48 kDa), whereas α1 + 2 forms 32 kDa dimers and 97 kDa octamers (theoretical ~26 kDa and 102 kDa). Differential refractive index (dRI) profiles are overlaid with fitted molecular weights. **i** α1-BRME1-MBD is a 33 kDa 2:2 complex (theoretical ~33 kDa), whereas α1 forms 17 kDa tetramers (theoretical ~20 kDa) and α2 forms 10 kDa monomers (theoretical ~ 9 kDa). **j** SAXS P(r) distributions of α1 + 2 + BRME1, α1 + 2 (dimer), α1 + 2 (octamer), and α2 (monomer) showing maximum dimensions (*Dmax*) of 190 Å, 160 Å, 200 Å, and 110 Å, respectively. **k** SAXS ab initio models generated from 30 independent DAMMIF runs with P1 symmetry. Scale bars: 50 Å. **l** Schematic of α1 + 2 dimer assembly into either a 2:2 complex with BRME1 or a higher-molecular weight assembly through self-association. Summary of SEC-SAXS data are shown in Supplementary Fig. 1g. Source data are provided as a Source Data file.

ternary complex (Fig. 2d, lane 7). The expression of BRME1-MBD and BRCA2-MBD was sufficient for the formation of the ternary complex with MEILB2 (Supplementary Fig. 2b), leading to the conclusion that MEILB2 mediates the BRCA2–BRME1 interaction through its C- and N-termini (Fig. 2a).

**Meiotic BRCA2 complex binds to SPATA22-MEIOB and RAD51.** To purify the in vivo complex, we performed BRME1 immunoprecipitation (IP) from mouse testis extracts followed by semi-quantitative mass spectrometry. Consistent with the in vitro data, we detected the specific enrichment of BRCA2 and MEILB2 in BRME1 IP compared with control IP, suggesting the presence of the ternary complex in vivo (Fig. 2e). The meiosis-specific ssDNA-binding complex SPATA22-MEIOB[9–11] was also enriched in the BRME1 IP (Fig. 2e). We confirmed these results by western blotting (Fig. 2f).

The reciprocal BRCA2 IP from testis extracts provided further evidence for the BRCA2-MEILB2-BRME1 ternary complex in vivo (Fig. 2g). Notably, we detected the enrichment of RAD51, but not DMC1, in BRCA2 IP (Fig. 2g), which was also the case for BRME1 IP (Fig. 2f). In contrast to the well-established direct interaction between BRCA2 and RAD51, there have only been in vitro studies characterizing the BRCA2-DMC1 interaction[12,13], and our in vivo data suggested that the physiological interaction between BRCA2 and DMC1 is weak or transient. Taken together, we conclude that BRCA2 forms a meiosis-specific complex with MEILB2 and BRME1 that interacts with the ssDNA-binding complex SPATA22-MEIOB and the recombinase RAD51, but not DMC1 (Fig. 2h).

**BRME1 is a downstream factor of MEILB2.** Our in vitro and in vivo biochemical experiments strongly suggest that BRME1 is a regulator of meiotic HR along with BRCA2-MEILB2. To test this, we investigated the localization of BRME1 by immunostaining of mouse spermatocytes. As expected, we detected punctate foci along the chromosome axes, colocalizing with the DSB marker RPA2 (Fig. 3a and Supplementary Fig. 3a). Double staining of BRME1 and MEILB2 showed that they largely colocalized from leptotene to early-pachytene (Fig. 3b). We also expressed GFP-tagged BRME1 truncations in spermatocytes by in vivo electroporation and examined their localizations (Fig. 3c). Punctate GFP signals were seen in the full-length, C-terminal, and MBD fragment of BRME1 (Fig. 3d, e and Supplementary Fig. 3b). In contrast, the N-terminal fragment was diffuse within the nucleus and failed to form punctate signals (Fig. 3d), whereas the protein expression was comparable to that of full-length protein (Supplementary Fig. 3c). These results suggest that the MBD of

BRME1 is the minimum region capable of recombination nodule localization.

To further investigate the localization mechanism, we examined BRME1 localization in *Meilb2*−/− spermatocytes along with *Spo11*−/− and *Dmc1*−/− controls. *Spo11*−/− mice fail to form DSBs[14], and thus BRME1 and RPA2 foci were totally absent (Fig. 3f). In *Dmc1*−/− mice where the induction of DSBs and localization of MEILB2 are not affected[7,15], we detected both RPA2 and BRME1 localization similar to WT (Fig. 3f). In *Meilb2*−/− mice, however, BRME1 foci were totally absent, whereas RPA2 foci were intact (Fig. 3f and Supplementary Fig. 3d). This indicates that BRME1 localization on meiotic recombination nodules depends on MEILB2.

**MEILB2-BRME1 is recruited by SPATA22-MEIOB.** Because MEILB2-BRME1 interacted with the ssDNA-binding complex SPATA22-MEIOB in IP experiments (Fig. 2e, f), we investigated MEILB2-BRME1 localization in *Meiob*−/− spermatocytes in which both SPATA22 and MEIOB localization is abolished[9]. Both MEILB2 and BRME1 foci were detectable but were significantly weaker in *Meiob*−/− spermatocytes compared with WT (Fig. 3g, h and Supplementary Fig. 3e), suggesting that SPATA22-MEIOB is not necessary for, but facilitates, the localization of MEILB2-BRME1 on ssDNA. Our data also suggest that the reduced recombinase localization in *Meiob*−/− and *Spata22*−/− reported in previous studies[10,11] is most likely to be a consequence of reduced MEILB2-BRME1 localization in these mutants.

**BRME1 stabilizes MEILB2 in vivo.** BRME1 is a downstream factor of MEILB2 in meiotic HR, but its function is unknown. To investigate the in vivo functions, we generated *Brme1*−/− mice by targeting exons 7 and 8 of *Brme1*, which encode the MBD (Fig. 4a). We obtained an allele in which 386 base pairs, including the whole of exon 7 and half of exon 8, were deleted, which also caused a frameshift (Fig. 4a). *Brme1*−/− spermatocytes completely lacked the punctate localization of BRME1 (Fig. 4b), suggesting that this was a loss-of-function allele.

Although *Brme1*−/− mice showed no apparent somatic phenotypes, *Brme1*−/− males, but not females, were infertile similar to *Meilb2*−/− mice[7]. Testes from *Brme1*−/− mice were smaller than WT (Fig. 4c), and spermatids were absent (Fig. 4d). Mature sperm were also completely absent in *Brme1*−/− epididymis (Fig. 4e). TUNEL staining showed massive cell death in prophase I spermatocytes in *Brme1*−/− testes, suggesting that the defects in spermatogenesis led to the compete loss of male gametes (Fig. 4f).

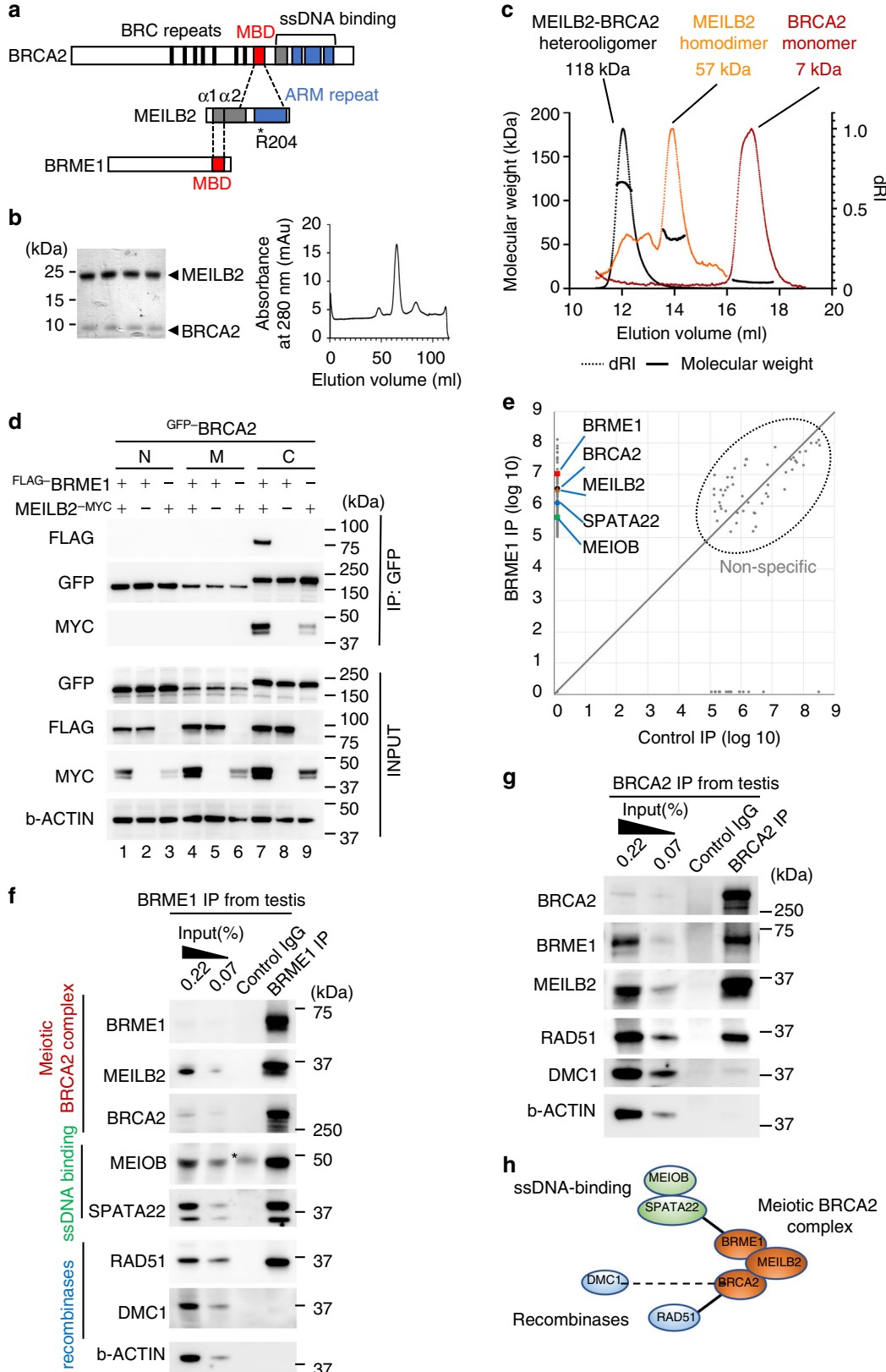

Our in vitro biochemistry experiments suggested that BRME1 functions as a stabilizer of MEILB2 by preventing its self-association (Fig. 1). To test its roles in vivo, we examined the localization of MEILB2 in $Brme1^{-/-}$ spermatocytes. As expected, we no longer detected any bright MEILB2 foci in $Brme1^{-/-}$ spermatocytes (Fig. 4g), and we saw 67% and 57% reductions in

MEILB2 signal intensity and foci number, respectively, compared with WT (Fig. 4h, i). These faint foci still resided on chromosome axes (Fig. 4h) and colocalized with RPA2 (Supplementary Fig. 4a) as in WT spermatocytes, suggesting that these faint foci were not background signals. Western blotting of testes extracts showed that the protein level of MEILB2 was also decreased in $Brme1^{-/-}$

**Fig. 2 Characterization of the BRCA2-MEILB2-BRME1 ternary complex. a** Schematic of interactions between BRCA2, MEILB2, and BRME1. MBD; MEILB2-binding domain, ARM repeat; armadillo-repeat domain. **b** Coomassie-stained gel of the heterocomplex of MEILB2 (a.a. 87–338; 27.8 kDa) and BRCA2 (a.a. 2219–2285; 10.3 kDa). The peak fractions from the size-exclusion chromatography (right graph) are shown. Purification of the individual proteins is shown in Supplementary Fig. 2a. **c** SEC-MALS analysis of MEILB2 (a.a. 87–338), BRCA2 (a.a. 2219–2285), and their heterocomplex. Differential refractive index (dRI) profiles are overlaid with fitted molecular weights. **d** IP with the GFP antibody from B16-F1 cells expressing [FLAG]-BRME1 and MEILB2-[MYC] and [GFP]-BRCA2 truncations; N (a.a. 1–981), M (a.a. 982–2035), and C (a.a. 2036–3329). **e** Quantitative mass spectrometry of BRME1 IP (vertical axis) and control IgG IP (horizontal axis). Combined peptide intensities are plotted for each protein. Genes involved in meiotic HR regulation are indicated. Note that the other DNA repair factors, such as RAD51, were not detected by the mass spectrometry analysis, likely owing to the limited sensitivity. **f, g** IPs from mouse testis extracts with the BRME1 (F) or BRCA2 (G) antibodies. Asterisk: IgG heavy chain. **h** Protein interaction map, identified by mass spectrometry and western blotting.

mice (Supplementary Fig. 4b). Taken together, we conclude that BRME1 functions as a stabilizer of MEILB2 in vivo.

**Defective localization of BRCA2 and recombinases.** We previously showed that electroporated [GFP]-BRCA2-MBD forms punctate foci at meiotic DSBs in a manner dependent on MEILB2[7]. Consistent with the reduction of MEILB2, the localization of [GFP]-BRCA2-MBD was also significantly impaired in Brme1[−/−] spermatocytes, although we could still detect faint axis-associated [GFP]-BRCA2-MBD foci in Brme1[−/−] mice after intensifying the signals (Fig. 4j).

The foci number of both RAD51 and DMC1 recombinases, which function downstream of BRCA2, was also significantly reduced in Brme1[−/−] spermatocytes from zygotene to early-pachytene, whereas the defects were milder than in Meilb2[−/−] spermatocytes where recombinases were almost totally delocalized (Fig. 4k, l and Supplementary Fig. 4c, d). This suggests that BRME1 is needed for the full accumulation of MEILB2-BRCA2 complexes and recombinases at DSBs.

We previously reported that the signal intensity of SPATA22 foci is significantly stronger in Meilb2[−/−] spermatocytes compared with WT[7]. Consistent with this, the SPATA22 foci were also stronger in Brme1[−/−] mice compared with WT, although the degree of increase was less than in Meilb2[−/−] spermatocytes (Supplementary Fig. 4e). Together, these data suggest a model in which MEILB2-BRME1 is initially clamped on ssDNA by SPATA22-MEIOB and then inhibits the excess loading of SPATA22-MEIOB likely by facilitating the loading of recombinases (Fig. 4m).

**Persistent DSBs reactivate the somatic-like DDR pathway.** Because recombinases are indispensable for homology-directed repair of DSBs, we investigated the progression of meiotic DSB repair in Brme1[−/−] spermatocytes. Consistent with the attenuated recombinase localization, unrepaired DSBs persisted until late prophase I (diplotene) as seen by the retention of RPA2 foci (Fig. 5a and Supplementary Fig. 5). The staining of another DNA-damage marker, the phosphorylated serine 139 of histone H2AX (γH2AX) that is normally restricted to sex chromosomes after pachytene, also persisted on the chromosomal axis until diplotene in Brme1[−/−] spermatocytes (Fig. 5b). Both RPA2 foci and the γH2AX signal gradually decreased toward the diplotene stage, suggesting that the residual DSBs were repaired in late prophase I by an alternative pathway that does not require MEILB2-BRME1.

Although fewer RAD51 foci were seen from zygotene until early-pachytene in Brme1[−/−] spermatocytes compared with WT (Fig. 4l) and tended to be limited to sex chromosomes in early pachytene (Fig. 5c), RAD51 foci reaccumulated in late-pachytene Brme1[−/−] spermatocytes (Fig. 5c). In contrast, the localization of DMC1 totally disappeared after the early-pachytene stage in Brme1[−/−] spermatocytes and never reaccumulated (Fig. 5d). This phenomenon likely reflects the reactivation of the somatic-like

DNA-damage response (DDR) in late-pachytene that involves RAD51 but not DMC1[16].

The triple staining of RAD51, γH2AX, and SYCP3 further confirmed that RAD51 foci did not colocalize with the γH2AX-marked DSB sites in early pachytene (except on the sex chromosomes) but reaccumulated globally at the residual DSB sites in Brme1[−/−] late-pachytene spermatocytes (Fig. 5e, f), whereas DMC1 foci did not (Fig. 5g). These results suggest that without BRME1 unrepaired DSBs persist until late prophase I due to the misloading of meiotic recombinases, which leads to the reactivation of somatic-like DDR in the late-pachytene stage (Fig. 5h).

**Synapsis and crossover formation defects.** Because strand invasion during DSB repair is indispensable for achieving homologous synapsis, we investigated the synapsis progression in Brme1[−/−] spermatocytes by staining for synaptonemal complex central element protein 3 (SYCE3)[17] (Supplementary Fig. 6). Compared with WT, there was significant accumulation of early prophase I (leptotene and zygotene) and reduction of late prophase I (diplotene and metaphase I) populations in Brme1[−/−] testes, suggesting that prophase I progression (i.e., synapsis progression) was delayed (Fig. 6a). A majority (80%) of Brme1[−/−] zygotene spermatocytes showed the partner switch phenotype, an indicator of non-homologous synapsis, suggesting that homologous synapsis is defective in Brme1[−/−] spermatocytes (Fig. 6b).

Even in pachytene spermatocytes, where most of the chromosomes completed synapsis, we could see a variety of synapsis defects in Brme1[−/−] spermatocytes. In WT spermatocytes, the X and Y chromosomes synapse within a narrow homologous region called the pseudoautosomal region (PAR) near the chromosome ends[18,19]. However, in Brme1[−/−] testes, a large portion of the spermatocytes had completely separated sex chromosomes (asynapsis) (Fig. 6c). Interestingly, we also observed excessive synapsis of X and Y chromosomes in Brme1[−/−] testes, where almost the entire Y chromosome axis was synapsed with the X chromosome (Fig. 6c). Because there is no sequence homology between the X and Y chromosomes except for the PAR, the excessive synapsis represents the occurrence of abnormal non-homologous synapsis at regions other than the PAR.

Even though sex chromosomes are the most susceptible to asynapsis, we also observed asynapsis between short autosomes in Brme1[−/−] spermatocytes (Fig. 6d). Further, even when autosomes were aligned, a gap was frequently observed in Brme1[−/−] spermatocytes (referred to as juxtaposition in Fig. 6d). This suggests that some autosomes managed to pair but failed to complete synapsis.

Consistent with the notion that DSB repair and subsequent homologous synapsis are a prerequisite for crossover formation, we found a significant reduction in foci number of the type I crossover marker MutL homolog 1 (MLH1)[20] in Brme1[−/−] spermatocytes compared with WT (Fig. 7a). The average MLH1 foci number in Brme1[−/−] spermatocytes was 17 (lower than the number of homologous pairs) implying that type I crossover

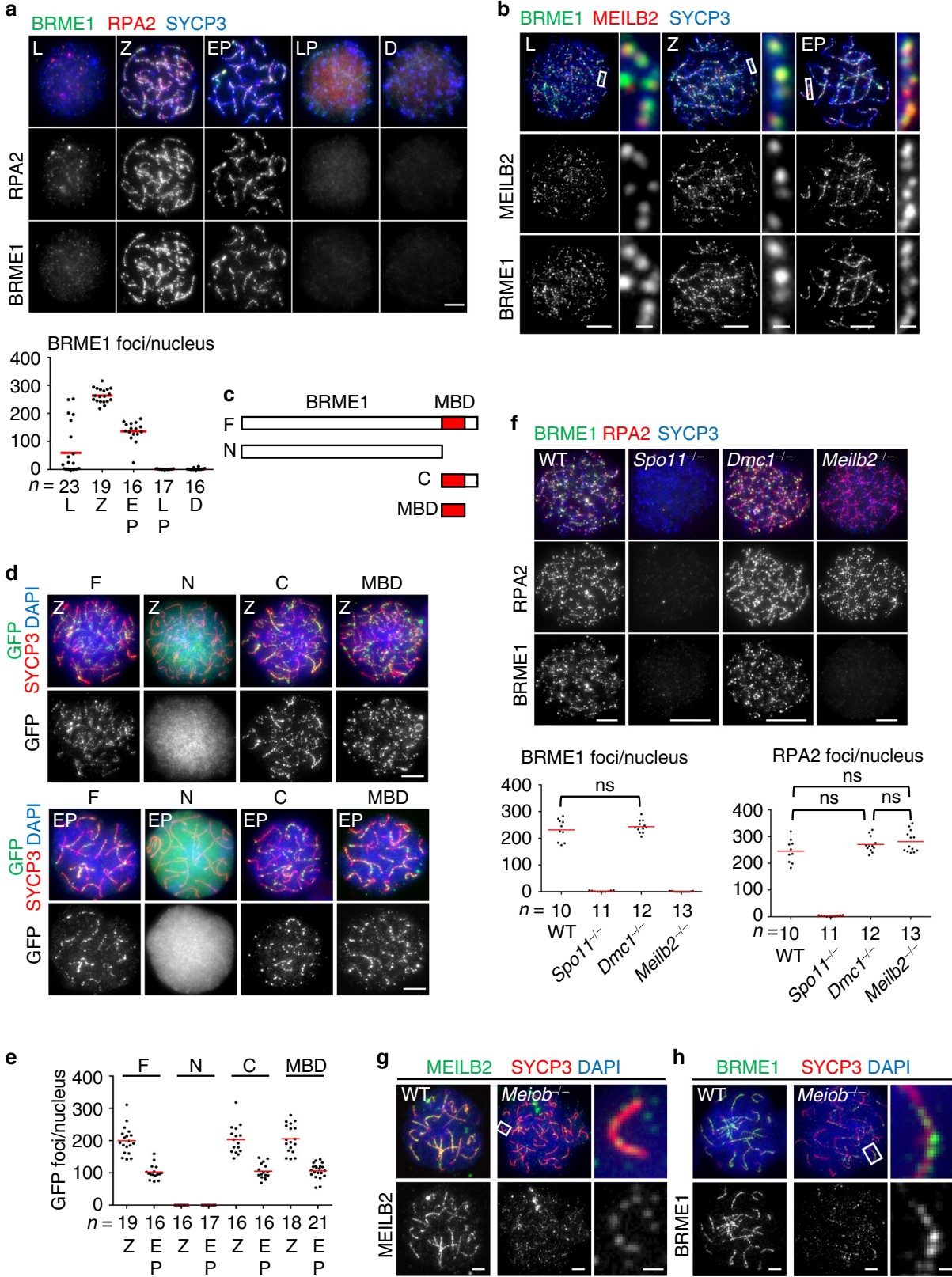

formation was abrogated in some of the homologous pairs. Indeed, *Brme1*^−/− metaphase I cells had univalent chromosomes that failed to form chiasmata/crossovers (Fig. 7b). Both sex chromosomes and short autosomes were observed to be univalent (Fig. 7b), consistent with the synapsis defects prevalent in both chromosomes (Fig. 6c, d).

The chiasmata are needed for the proper alignment of homologous chromosomes in metaphase I by counteracting the pulling forces exerted by the spindle microtubules[21,22]. In line with this, we observed a number of metaphase I cells with misaligned univalent chromosomes in *Brme1*^−/− testes (Fig. 7c). Further, a TUNEL assay showed significant cell death in

**Fig. 3 BRME1 is a meiotic DSB-associating protein. a** Immunostaining of WT spermatocytes. Graph: the number of BRME1 foci associated with the chromosome axes. Red bars: mean value. *n* shows the analyzed spermatocyte number pooled from three mice. **b** Immunostaining of WT spermatocytes. 79%, 81%, and 89% of the BRME1 foci stained positive for MEILB2 and 83%, 85%, and 86% of the MEILB2 foci stained positive for BRME1 at the leptotene (five cells), zygotene (nine cells), and early-pachytene (seven cells) stages, respectively. Cells were pooled from two mice. **c** Schematic of BRME1 truncations. F (a.a. 1–605), N (a.a. 1–518), C (a.a. 519–605), and MBD (a.a. 519–600). **d** Immunostaining of WT spermatocytes in zygotene (Z; top) or early-pachytene (EP; bottom) expressing ^GFP^-BRME1 truncations shown in **c**. **e** The number of ^GFP^-BRME1 foci associated with the chromosome axes. Red bars: mean value. *n* shows the analyzed spermatocyte number pooled from three electroporated mice for each transgene. **f** Immunostaining of zygotene spermatocytes. Intensified BRME1 signals in *Meilb2*^−/−^ are shown in Supplementary Fig. 3d. Graph: the number of BRME1 (left) or RPA2 (right) foci associated with the chromosome axes. Red bars: mean value. *n* shows the analyzed spermatocyte number pooled from two mice for each genotype. **g**, **h** Immunostaining of zygotene spermatocytes. MEILB2 **g** and BRME1 **h** are stained. Leptotene (L), zygotene (Z), early-pachytene (EP), late-pachytene (LP), diplotene (D). All analyses were with two-tailed *t* tests. ns, not significant. Scale bars: 5 µm or 1 µm (magnified panel). Source data are provided as a Source Data file.

metaphase I spermatocytes in seminiferous tubules at stage XII (Fig. 7d). Thus, there are two rounds of cell death in *Brme1*^−/−^ testes—the first in pachytene spermatocytes at stage V–VI (Fig. 4f) and the second in metaphase I at stage XII (Fig. 7d). Together these results lead to the conclusion that BRME1 is indispensable for the DSB repair, homologous synapsis, and crossover formation that are needed for progression past metaphase I (Fig. 7e).

**MEILB2 and BRME1 inhibit mitotic HR**. Expression of MEILB2 has been reported in mouse embryonic stem cells and human cancer cells, but the somatic role of MEILB2 has been largely unaddressed[8]. During our pulldown experiments, we found that transfected MEILB2 formed punctate foci within the mitotic nucleus that colocalized with endogenous RAD51 (Fig. 8a), suggesting that MEILB2 localizes to DSBs even in mitotic cells. We speculated that this might be mediated by interactions with endogenous BRCA2. To test this hypothesis, we used the MEILB2-R204T mutant protein. The pulldown assay confirmed that MEILB2-R204T almost totally abolished the interaction with both the C-terminus and MBD of BRCA2 (Fig. 8b). MEILB2-R204T no longer formed foci at DSBs (Fig. 8a), whereas the protein expression was comparable to WT protein (Supplementary Fig. 7a), proving that DSB localization of MEILB2 depends on BRCA2 binding. It is interesting to note that MEILB2-R204T still localized to meiotic DSBs in spermatocytes (Fig. 8c and Supplementary Fig. 7b). The difference is likely explained by the presence of a meiosis-specific SPATA22-MEIOB-dependent recruitment pathway (Fig. 4m).

In contrast to MEILB2, the transfected BRME1 did not localize to mitotic DSBs and instead formed a diffuse nuclear signal (Fig. 8d). However, when we co-expressed BRME1 with MEILB2, BRME1 formed punctate foci colocalizing with RAD51 (Fig. 8d). The MBD fragment of BRME1 also exhibited MEILB2-dependent localization at mitotic DSBs (Supplementary Fig. 7c), leading to the conclusion that MEILB2 and BRME1 form complexes and localize at mitotic DSBs in a BRCA2-dependent manner.

*BRME1* is upregulated in a number of human cancer tissues, and its cancer type-specific expression pattern is reminiscent of that of *MEILB2*[8] (Supplementary Fig. 7d). This implies the oncogenic function of *MEILB2* and *BRME1*, which is likely exerted by the disruption of the mitotic HR pathway. To test this, we treated cultured somatic mouse cells (B16-F1) with mitomycin C (MMC), which artificially induces DNA damage and activates mitotic HR. In mock-treated cells, the increase in RAD51 foci after MMC treatment suggested that HR was activated (Fig. 8e). However, when we overexpressed either MEILB2 or BRME1, the elevation of RAD51 foci formation was largely abrogated (Fig. 8e). The expression of MEILB2-R204T did not inhibit the formation of RAD51 foci, suggesting that the inhibition of mitotic HR by MEILB2 relies on BRCA2 binding (Fig. 8e).

To rule out the possibility that the inhibitory effect might be an artifact of our MMC system or might be cell-type specific, we used heterologous human U2OS cells with an inducible DSB reporter[23]. With the addition of 4-OHT and Shield-1, FokI is expressed and generates DSBs at the LacO repeat, which is manifested as a single DSB focus per nucleus (Fig. 8f). After induction of DSBs in the mock-treated cells, there was robust accumulation of RAD51 at the single DSB site (Fig. 8g). However, if we overexpressed either MEILB2 or BRME1, the accumulation of RAD51 at the DSB was largely impaired. Thus the meiotic BRCA2 complex, which has central roles in meiotic HR, is an inhibitory factor for mitotic HR when ectopically expressed in somatic cells. This inhibition is, at least in the case of MEILB2, caused by its ability to bind to endogenous BRCA2, which may sequester BRCA2 from functional complexes or modulate BRCA2 into non-functional structures[24,25].

## Discussion

We have identified BRME1 as a subunit of the meiotic BRCA2-MEILB2-BRME1 complex, which binds to the SPATA22-MEIOB ssDNA-binding complex and to the recombinase in vivo. Similar to MEILB2 and BRCA2[7,26], BRME1 functions as a recruiter of meiotic recombinases onto ssDNA. The primary function of BRME1 seems to be the stabilizer of MEILB2 by binding to α1 of MEILB2 and preventing MEILB2 self-association. Given that BRME1 localization at DSBs depends on MEILB2, we conclude that BRME1 is a downstream factor of MEILB2 and plays a role in the stabilization of MEILB2 at meiotic DSBs.

The delayed apoptosis in *Brme1*^−/−^ spermatocytes compared with *Meilb2*^−/−^ spermatocytes allowed us to investigate the consequence of a defective homology search in late prophase I. The *Brme1*^−/−^ pachytene spermatocytes showed a variety of synapsis defects, including autosomal asynapsis, sex chromosome asynapsis, and, paradoxically, sex chromosome excessive synapsis. Because the DNA homology search during DSB repair is a prerequisite for the progression of homologous synapsis, asynapsis is seen in a variety of recombination mutant mice[14,15,27,28]. However, the excessive synapsis of sex chromosomes seen in *Brme1*^−/−^ spermatocytes is a paradoxical phenotype so far never reported. Given there is no sequence homology between X and Y chromosomes other than the PAR, the excessive synapsis is likely to be a consequence of a defective homology search and subsequent non-homologous synapsis occurring outside the PAR on sex chromosomes. In addition to the synapsis defects, late prophase I *Brme1*^−/−^ spermatocytes showed unrepaired DSBs that reactivated mitosis-like DDR. This pathway involves only RAD51 and not DMC1 recombinase, and meiosis-specific MEILB2-BRME1 no longer seems to be required. Similar reactivation of mitosis-like DDR was recently reported in late-pachytene spermatocytes exposed to gamma irradiation[16].

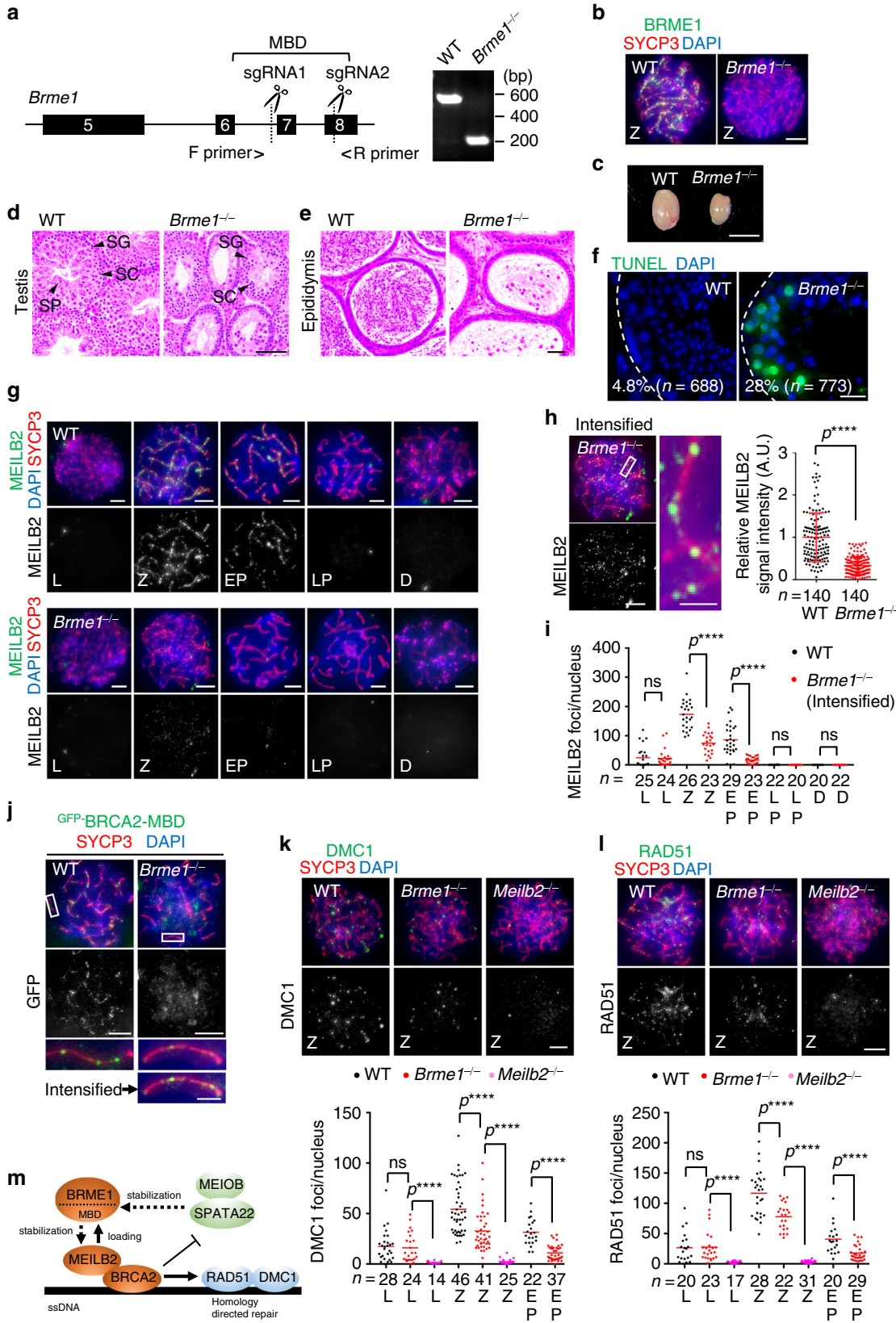

Mitotic BRCA2 forms stable homodimers in vitro, and each BRCA2 protomer binds to four or five RAD51 molecules and facilitates unidirectional 3′–5′ growth of RAD51 protofilaments on the ssDNA[29]. In contrast to this somatic BRCA2 model, our purification and analysis of the BRCA2-MEILB2-BRME1 complex suggest that the meiotic BRCA2 complex forms a stable hetero-oligomeric complex in vitro. This raises the question of whether this non-canonical BRCA2 complex can bind to RAD51, as did the homodimer. We provide in vivo evidence showing that both BRCA2 and BRME1 IPs enrich RAD51 from testes extracts,

**Fig. 4 BRME1 is a stabilizer of MEILB2-BRCA2. a** Schematic of the *Brme1* knockout allele. Rectangle; exon. The two sgRNA-targeting sites and the location of the primers used for genotyping are shown. The gel shows the PCR results for WT and *Brme1*$^{-/-}$ alleles. **b** Immunostaining of zygotene spermatocytes. Scale bar: 5 μm. **c** Testes from WT and *Brme1*$^{-/-}$ males. Scale bar: 5 mm. Testis **d** or epididymis **e** sections from 8-week-old WT and *Brme1*$^{-/-}$ males stained with hematoxylin and eosin. Spermatogonia (SG), spermatocyte (SC), and spermatid (SP). Scale bar: 100 μm. **f** Testis sections with stage V–VI seminiferous tubules from 8-week-old WT and *Brme1*$^{-/-}$ males stained with TUNEL and DAPI. The percentages of TUNEL-positive seminiferous tubules (those containing more than three TUNEL-positive cells) were quantified. *n* shows the analyzed seminiferous tubule number pooled from two mice for each genotype. Scale bar: 15 μm. **g** Immunostaining of spermatocytes. Scale bar: 5 μm. **h** Immunostaining of *Brme1*$^{-/-}$ zygotene spermatocyte. Graph: the quantification of MEILB2 foci intensities in zygotene spermatocytes, normalized to the average value of WT. Red bars: mean value with SD. *n* shows the analyzed foci number pooled from seven cells from two mice for each genotype. Scale bars: 5 μm or 1 μm in the magnified panel. **i** The number of MEILB2 foci associated with the chromosome axes. Red bars: mean value. *n* shows the analyzed spermatocyte number pooled from three mice for each genotype. **j** Immunostaining of zygotene spermatocytes expressing $^{GFP}$-BRCA2-MBD (a.a. 2117–2339). Scale bars: 5 μm or 1 μm in the magnified panel. **k, l** Immunostaining of zygotene spermatocytes. DMC1 **k** or RAD51 **l** were stained. The images in the other stages are shown in Supplementary Fig. 4c, d. Graph: the number of foci associated with the chromosome axes. Red bars: mean value. *n* shows the analyzed spermatocyte number pooled from three mice for each genotype. Scale bar: 5 μm. **m** Schematic of meiotic BRCA2 complex assembly pathways. All analyses used two-tailed *t* tests. ns: not significant. ****$p < 0.0001$. Leptotene (L), zygotene (Z), early-pachytene (EP), late-pachytene (LP), diplotene (D). Source data are provided as a Source Data file.

suggesting that the BRCA2-MEILB2 heterocomplex binds to RAD51. This specialized meiotic BRCA2 complex might allow for the loading of DMC1, the meiosis-specific paralog of RAD51, onto ssDNA by removing SPATA22-MEIOB or might have some other unknown meiosis-specific functions. Although we did not detect a physical protein interaction between meiotic BRCA2 complex with DMC1 in vivo, DMC1 localization was totally abolished in *Meilb2*$^{-/-}$ and significantly decreased in *Brme1*$^{-/-}$ spermatocytes. This might be an indirect consequence of RAD51 mislocalization in these mutants, as suggested by a yeast study where RAD51 facilitates the localization of DMC1[30], or there might be direct regulation of DMC1 by BRCA2-MEILB2-BRME1, whereas their physical interactions are weak or transient.

We showed that ectopic overexpression of MEILB2 and BRME1 in somatic cells inhibits the mitotic HR pathway. This inhibition by MEILB2 is caused by its ability to bind to endogenous BRCA2, as shown by the loss of inhibitory effect after the expression of MEILB2-R204T. BRME1 may also exert its inhibitory function by binding to some mitotic DDR proteins, and it should be a goal for future studies to identify the responsible BRME1 partner proteins. Several previous studies reported similar dominant-negative inhibition of HR. For example, overexpression of the TR2 fragment of BRCA2, which binds to RAD51, showed a reduction in HR activation in human cells by binding and sequestering RAD51 from endogenous BRCA2[31]. Further, the meiosis-specific protein SYCP3 binds to BRCA2 and inhibits mitotic HR[25]. EMSY is another example, which binds to the N-terminus of BRCA2 and, when sporadically overexpressed, inhibits BRCA2 function by crippling the BRCA2-RAD51 complex at DSBs[24]. *SYCP3* is overexpressed in various human breast cancer cell lines, and *EMSY* is amplified almost exclusively in sporadic breast and ovarian cancers[32,33], suggesting that overexpression of *SYCP3* and *EMSY* phenocopies cancer related-*BRCA2* mutations. Given that *MEILB2* and *BRME1* are also aberrantly expressed in human cancer cell lines, the amplification of gene products involved in BRCA2 function and subsequent hijacking of the intrinsic HR pathway might be a driving force in the development of certain sporadic cancers.

## Methods

**Mice.** Knockout mice for *Spo11*, *Dmc1*, *Meiob*, and *Meilb2* were reported earlier[7,9,10,14,15]. *Brme1*$^{-/-}$ mice were generated in this study. All WT and knockout mice were congenic with the C57BL/6J background. All animal experiments were approved by the Institutional Animal Care and Use Committee (#1316/18).

**Preparation of sgRNA for *Brme1*$^{-/-}$ mice.** For the generation of sgRNA, 18–20 bp primers were annealed and inserted into the BsaI sites of a pUC57-sgRNA expression vector. The pUC57-sgRNA expression vector was linearized with the

DraI enzyme, and in vitro transcription was performed with a MEGAshortscript kit (Ambion; AM1354).

**Generation of *Brme1*$^{-/-}$ mice.** Mouse zygotes were collected from superovulated 3–4-week-old C57BL/6J females (Janvier Labs) mated with C57BL/6J males (Janvier Labs) overnight. SpCas9 protein (30 ng/μl; Sigma) and sgRNA (12.5 ng/μl) were mixed to form ribonucleoprotein complexes and injected into zygotes in M2 medium. Injected zygotes were allowed to recover for 2–4 h in KSOM medium in a humidified $CO_2$ incubator at 37°C and then were transferred into pseudopregnant 8–20-week-old SWISS female mice (Janvier Labs). For identification of founders, ear punch biopsies were subjected to DNA-extraction procedures, and the extracted DNA was amplified with primers flanking the sgRNA target site. PCR products from each founder were sequenced, and the mutated founder mice were crossed with C57BL/6J WT mice to avoid potential off-target mutations. The *Brme1* allele was genotyped using the following primers: Forward; 5′-TTCAGG GTAGGATAGGATGGGG-3′, Reverse; 5′-CTTGTAATCTGCTGCAGCCT-3′.

**Histological analysis.** Testes and epididymis were fixed in Bouin's fixative for 24 h at room temperature and embedded into paraffin blocks. Slices of 8 μm thickness were stained with hematoxylin and eosin. TUNEL analysis was carried out with an ApopTag Plus In Situ Apoptosis Fluorescein Detection Kit (S 7111; Millipore).

**Antibodies.** The following antibodies were used: rabbit antibodies against BRME1 (this study), BRCA2 (this study), MEILB2[7], GFP (Invitrogen; A11122), DMC1 (Santa Cruz Biotechnology; sc-22768), RAD51 (Thermo Fisher Scientific; PA5-27195), SPATA22 (Proteintech Group Inc; 16989-1-AP), SYCE3[7], and MEIOB (EMD Millipore; ABE1414); mouse antibodies against BRME1 (this study), DMC1[7], β-ACTIN (Sigma; A2228-100UL), MLH1 (BD Biosciences; 51-1327GR), γH2AX (EMD Millipore; 05-636), FLAG (Sigma; F1804-50UG), and MYC (MBL; M192-3); rat antibody against RPA2 (Cell signaling technology; 2208); sheep antibody against BRCA2[34]; and chicken antibody against SYCP3[7].

**Antibody production.** cDNA encoding *Brme1* (a.a. 1–200) and *Brca2* (a.a. 2730–3029) were cloned into the pET28c+ vector (Millipore). The HIS-tagged recombinant proteins were expressed in BL21 (DE3) cells, solubilized in a denaturing buffer (6 mM HCl-guanidine and 30 mM Tris-HCl (pH 7.5)), and purified with Ni-NTA resin (Qiagen). The recombinant proteins were dialyzed in phosphate-buffered saline (PBS) and used to immunize the animals. The polyclonal antibodies were affinity purified on antigen-coupled Sepharose beads (GE Healthcare).

**Reverse transcription PCR.** Total RNA was isolated from tissues using the RNeasy Mini kit (Qiagen). cDNAs were generated by iScript reverse transcription super mix (Bio-Rad), and PCR amplification was performed using standard DNA polymerase. The primers used were as follows: *Brme1*–forward; 5′-GCCCAGATT TCTCAGGGAAAGA-3′, *Brme1*–reverse; 5′-CCAACTGAGTTCTGGGAAGGA-3′, *Meilb2*–forward; 5′-GCCTGCCGGAACATGGA-3′, *Meilb2*–reverse; 5′-TGGT TTTGACGACCTCCTCG-3′, *Gapdh*–forward; 5′-TTCACCACCATGGAGAA GGC-3′, and *Gapdh*–reverse; 5′-GGCATGGACTGTGTGGTCATGA-3′.

**Exogenous protein expression in the testis.** Plasmid DNA was electroporated into live mouse testes by in vivo electroporation technique[35]. In brief, male mice at PD16–30 were anesthetized with pentobarbital, and the testes were pulled from the abdominal cavity. Plasmid DNA (10 μl of 5 μg/μl solution) was injected into each testis using glass capillaries under a stereomicroscope (M165C; Leica). Testes were held between a pair of tweezers-type electrodes (CUY21; BEX), and electric pulses

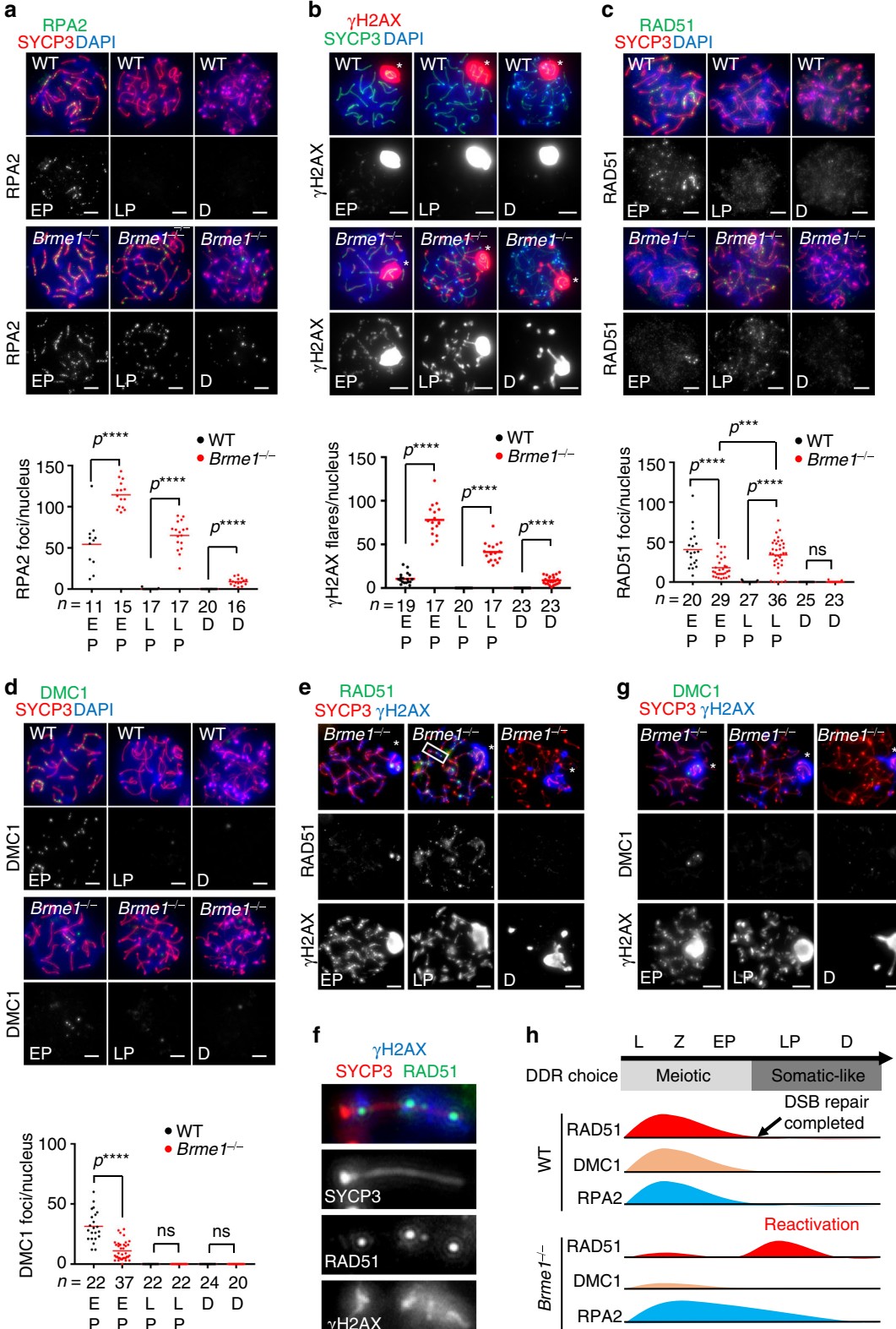

**Fig. 5 Unrepaired DSBs persist and activate somatic-like DDR in *Brme1*⁻/⁻ mice. a–d** Immunostaining of spermatocytes. RPA2 **a**, γH2AX **b**, RAD51 **c**, and DMC1 **d** were stained. Graph: the number of foci associated with the chromosome axes. Asterisks in **b**: sex chromosomes. Red bars: mean value. *n* shows the analyzed spermatocyte number pooled from three mice for each genotype. **e–g** Immunostaining of *Brme1*⁻/⁻ spermatocytes. RAD51 **e** and DMC1 **g** were stained. A chromosome in **e** is magnified in **f**. Asterisks: sex chromosomes. **h** Schematic of DDR pathway activation in WT and *Brme1*⁻/⁻ males. Unrepaired DSBs persist in *Brme1*⁻/⁻ males owing to the low activation of meiotic recombinases in early prophase I, which subsequently activates somatic-like DDR in late prophase I. All analyses used two-tailed *t*-tests. ns: not significant. ****$p < 0.0001$. Leptotene (L), zygotene (Z), early-pachytene (EP), late-pachytene (LP), diplotene (D). Scale bars: 5 μm or 1 μm in the magnified panel. Source data are provided as a Source Data file.

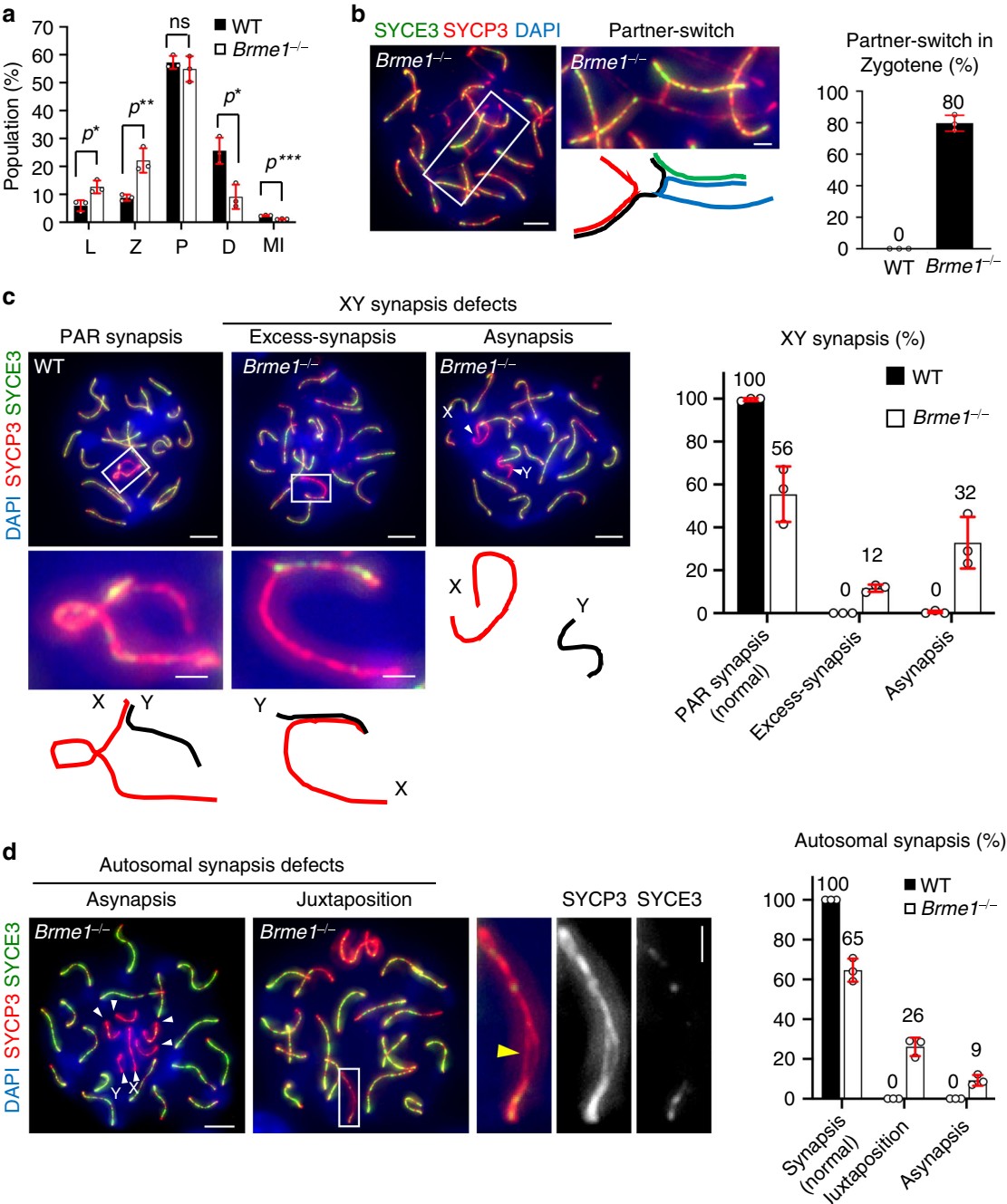

**Fig. 6 Synapsis defects in *Brme1*⁻/⁻ spermatocytes. a** Population of meiotic substages in WT (2623 cells) and *Brme1*⁻/⁻ (2565 cells) pooled from three mice for each genotype. The mean values of three independent experiments from three different mice are shown. Leptotene (L), zygotene (Z), pachytene (P), diplotene (D), metaphase I (MI). Error bars: SD. **b** Immunostaining of *Brme1*⁻/⁻ zygotene spermatocytes. Magnified pictures with schematics highlight the partner switch. Graph: the ratio of zygotene spermatocytes (74 cells pooled from three mice for each genotypes) showing at least one partner switch. The mean values of three independent experiments from three different mice are shown. Error bars: SD. Scale bars: 5 μm or 1 μm in the magnified panel. **c** Immunostaining of pachytene spermatocytes. Magnified pictures with schematics highlight the sex chromosomes. Arrowheads: sex chromosome asynapsis. Graph: the frequency of cells with sex chromosome synapsis defects in WT (191 cells) and *Brme1*⁻/⁻ (285 cells) pooled from three mice for each genotype. The mean values of three independent experiments from three different mice are shown. Error bars: SD. Scale bars: 5 μm or 1 μm in the magnified panel. **d** Immunostaining of pachytene spermatocytes. Arrowheads: asynapsis. The magnified picture highlights the incompletely synapsed autosomes (juxtaposition) with a gap between aligned homologs (yellow arrowhead). Graph: the frequency of autosomal synapsis defects in WT (192 cells) and *Brme1*⁻/⁻ (280 cells) pooled from three mice for each genotype. The mean values of three independent experiments from three different mice are shown. Error bars: SD. Scale bars: 5 μm or 1 μm in the magnified panel. All analyses used two-tailed *t* tests. ns: not significant. *$p < 0.05$, **$p < 0.01$, ***$p < 0.001$. Source data are provided as a Source Data file.

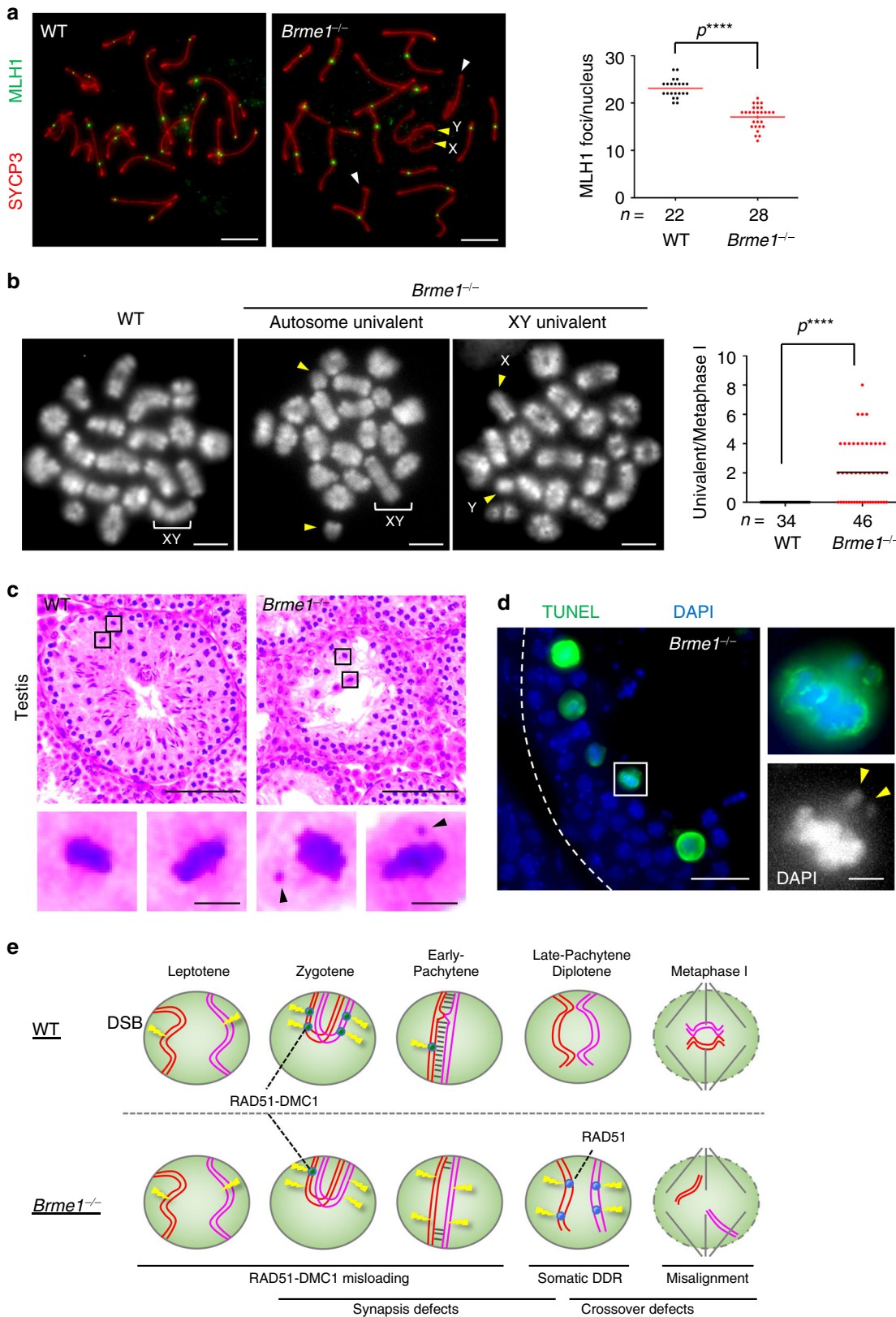

were applied four times and again four times in the reverse direction at 35 V for 50 ms for each pulse. The testes were then returned to the abdominal cavity, and the abdominal wall and skin were closed with sutures. The testes were removed 24 h after the electroporation, and immunostaining was performed.

**Immunostaining of spermatocytes**. Testis cell suspensions were prepared by mincing the tissue with flathead forceps in PBS, washing several times in PBS, and

resuspending in a 1:1 mixture of PBS and hypotonic buffer (30 mM Tris (pH 7.5), 17 mM trisodium citrate, 5 mM EDTA, 2.5 mM DTT, 0.5 mM PMSF, and 50 mM sucrose). After 10 min, the sample was centrifuged and the supernatant was aspirated. The pellet was resuspended in a 1:2 mixture of PBS and 100 mM sucrose. A total of 20 μl of fixation buffer (1% paraformaldehyde, 0.15% Triton X-100, and 1 mM Sodium Borate (pH 9.2 adjusted by NaOH)) was applied to a glass slide, and 3 μl of the cell suspension was added to the drop, allowed to fix for 1 h at room

**Fig. 7 Crossover formation defects in Brme1⁻/⁻ spermatocytes. a** Immunostaining of late-pachytene spermatocytes. Two autosomes and sex chromosomes without MLH1 foci are highlighted by white and yellow arrowheads, respectively. Graph: the number of MLH1 foci per spermatocyte associated with the chromosome axis. Red bars: mean value. n shows the analyzed cell number pooled from three mice. Scale bar: 5 μm. **b** Metaphase I spermatocyte spreads from WT and Brme1⁻/⁻ males stained with DAPI. Arrowheads: univalent chromosomes. Graph: the number of univalent chromosomes per metaphase I spermatocyte. The mean value is shown as a black bar. n shows the analyzed cell number pooled from three mice. Scale bar: 5 μm. **c** Testis sections from 8-week-old WT and Brme1⁻/⁻ males stained with hematoxylin and eosin. Arrowheads: the misaligned univalent chromosomes in metaphase I. Scale bars: 100 μm or 5 μm in the magnified panel. **d** Brme1⁻/⁻ testis sections with stage XII seminiferous tubule from 8-week-old male mice stained with TUNEL and DAPI. A TUNEL-positive metaphase I spermatocyte with misaligned univalent chromosomes in the magnified panel. Arrowheads: univalent chromosomes. Scale bars: 15 μm or 5 μm in the magnified panel. **e** Schematic summary of Brme1⁻/⁻ male phenotypes. All analyses used two-tailed t tests. ****p < 0.0001. Source data are provided as a Source Data file.

temperature, and air-dried. For immunostaining, the slides were incubated with primary antibodies in PBS containing 5% bovine serum albumin for 2 h and then with Alexa Fluor 488-, 594-, or 647-conjugated secondary antibodies (1:1000 dilution, Invitrogen) for 1 h at room temperature. The slides were washed with PBS and mounted with VECTASHIELD medium with DAPI (Vector Laboratories).

**Staging of meiotic prophase I**. In Fig. 6a and Supplementary Fig. 6, the meiotic prophase I spermatocytes are subdivided into stages by the staining of SYCP3 (chromosome axis) and SYCE3 (synaptonemal complex) as follows: leptotene (no SYCE3), zygotene (partially assembled SYCE3), pachytene (fully assembled SYCE3), diplotene (disassembled SYCE3), and metaphase I (SYCP3 accumulation at centromeres). In the other figures, the meiotic prophase I spermatocytes are subdivided into stages by the staining of SYCP3 alone as follows: leptotene (dotty or discontinuous SYCP3, without any synapsis), zygotene (discontinuous to continuous linear SYCP3, with partial synapsis), early-pachytene (linear SYCP3 with complete synapsis), late-pachytene (linear SYCP3 with complete synapsis with thickened SYCP3 ends and brighter XY axes), and diplotene (linear SYCP3 with desynapsis). Pachytene-like cells in the Brme1⁻/⁻ males frequently had both completely synapsed autosomes from end to end and a few chromosomes with complete asynapsis or juxtaposition.

**Metaphase I chromosome spreads**. Testis cell suspensions were prepared in PBS and washed several times in PBS. Cells were resuspended in hypotonic buffer (75 mM KCl) and kept for 10 min at room temperature. After hypotonic treatment, the cell suspensions were washed in Carnoy solution (10% acetic acid, 30% chloroform, and 60% ethanol) three times, placed on slides, and air-dried. The slides were mounted with VECTASHIELD medium with DAPI (Vector Laboratories).

**Preparation of testis extract and immunoprecipitation**. Testes were removed from male C57BL/6J mice and suspended in extraction buffer (20 mM Tris-HCl (pH 7.5), 50 mM KCl, 0.4 mM EDTA, 5 mM MgCl₂, 10% glycerol, 0.1% Triton X-100, and 1 mM β-mercaptoethanol) supplemented with cOmplete Protease Inhibitor (Roche) and Phosphatase Inhibitor (Roche). After homogenization, the cell extract was centrifuged at 50,000 × g for 30 min at 4°C and the supernatant (chromatin extract) was isolated. The extract was supplemented with Dynabeads protein A (Thermo Fisher Scientific) conjugated with 80 μg of BRME1, MEILB2, or BRCA2 antibodies or control IgG and incubated for 6 h at 4°C. The beads were washed with high-salt buffer (20 mM HEPES (pH 7.0), 400 mM KCl, 5 mM MgCl₂, 10% glycerol, 0.1% Triton X-100, and 1 mM β-mercaptoethanol) supplemented with cOmplete Protease Inhibitor (Roche) and Phosphatase Inhibitor (Roche). The samples were eluted with 0.1 M glycine (pH 2.5).

**Mass spectrometry analysis**. The eluted samples were processed according to the modified filter-aided sample preparation method[36]. In brief, the protein samples were reduced with DTT, alkylated with methyl methanethiosulfonate, and digested using trypsin on Nanosep 30k Omega centrifugation filters (Pall Life Sciences). The digested peptides were treated using the HiPPR detergent removal resin kit (PN 88305, Thermo Fisher Scientific, Waltham, MA, USA), and the eluted peptides were further purified using Pierce peptide desalting spin columns (PN 89851, Thermo Fisher Scientific), both according to the manufacturer's instructions.

The purified peptide samples were analyzed on a Q Exactive HF mass spectrometer interfaced with an Easy-nLC1200 nanoflow liquid chromatography system (Thermo Fisher Scientific). Peptides were trapped on an Acclaim Pepmap 100 C18 trap column (Thermo Fischer Scientific) and separated on an analytical column packed in-house with Reprosil-Pur C18 material (3 μm, Dr. Maisch, Germany) using a 60 min gradient from 0.2% formic acid to 80% acetonitrile and 0.2% formic acid at a flow rate of 300 nL/min. The mass spectrometer was operating in data-dependent mode, with the precursor ion spectra followed by the HCD spectra of the 10 most abundant precursors at a collision energy of 28 and a dynamic exclusion duration of 20 s.

Peptide identification and quantification were performed using Proteome Discoverer version 2.2 (Thermo Fisher Scientific). The LC-MS data were matched against the combined SwissProt and TrEMBL Mus musculus database using Mascot 2.5.1 (Matrix Science, London, United Kingdom) as a database search engine with trypsin and one allowed missed cleavage as an enzyme rule, with a precursor tolerance of 5 ppm and fragment tolerance of 0.02 Da, and with methionine oxidation set as a variable modification and methylthiolation on cysteine set as a fixed modification. The fixed-value PSM validator was used to assess the quality of the peptide matches. Precursor ion quantification was accomplished via the Minora feature detection node in Proteome Discoverer 2.2, with the maximum peak intensity values of the unique peptides used for protein quantification.

**Microscopy**. Images were obtained on a microscope (Olympus IL-X71 Delta Vision; Applied Precision) equipped with ×100 NA 1.40 and x60 NA 1.42 objectives, a camera (CoolSNAP HQ; Photometrics), and softWoRx 5.5.5 acquisition software (Delta Vision). Acquired images were processed with Photoshop (Adobe).

**Y2H assay**. Y2H screening was performed by Hybrigenics Services, Paris, France. The coding sequence for Meilb2 was cloned into pB27, and the construct was used as a bait to screen a random-primed mouse testis cDNA library constructed in pP6. Using a mating approach with YHGX13 and L40ΔGal4 yeast strains, 75 million clones (sevenfold the complexity of the library) were screened. Positive colonies were selected on a medium lacking tryptophan, leucine, and histidine and supplemented with 50 mM 3-aminotriazole. The prey fragments of the positive clones were amplified by PCR and sequenced. The resulting sequences were used to identify the corresponding interacting proteins in the GenBank database (NCBI) using a fully automated procedure. Genes that were identified more than one time (more than one original clone) were considered to be a positive interactor. For the Y2H assay, Meilb2 and Brca2-C (a.a. 2036–3329) cDNAs were cloned into the pGBKT7 vector. Brme1 cDNAs were cloned into the pGADT7 vector. These bait and prey were co-transformed into the yeast strain AH109, and the positive transformants were selected on nutrition-restricted plates (SD–tryptophan–leucine–histidine–adenine).

**Cell culture**. Cell line B16-F1 and U2OS-DSB reporter cell line[23] were maintained in DMEM (GIBCO Life Technologies) supplemented with 10% FBS (Invitrogen), 100 U/ml Penicillin–Streptomycin (GIBCO Life Technologies), and 2.5 μg/ml Plasmocin (InvivoGen) in a humidified atmosphere of 5% CO₂ at 37 °C. Transfection was performed using Lipofectamine 2000 transfection reagent (Invitrogen) and Optimem (GIBCO Life Technologies). For the staining of the B16-F1 cell line, cells were treated with extraction buffer (20 mM HEPES-KOH (pH 7.9), 50 mM NaCl, 3 mM MgCl₂, 300 mM sucrose, 0.5% Triton X-100) and fixed with 4% formaldehyde in PBS. For the staining of the U2OS-DSB reporter cell line, cells were fixed with 4% formaldehyde in PBS and treated with extraction buffer (0.2% Triton X-100 in PBS) for 15 min. For the induction of DSBs in the B16-F1 cell line, cells were treated overnight with 10 μl MMC (Sigma). For the DSB induction in the U2OS-DSB reporter cell line, cells were treated for 8 h with 1 μM 4-OHT and 1 μM Shield-1.

**Pulldown assay**. Transfected B16-F1 cells were suspended in extraction buffer (20 mM Tris-HCl (pH 7.5), 50 mM KCl, 0.4 mM EDTA, 5 mM MgCl₂, 10% glycerol, 0.1% Triton X-100, and 1 mM β-mercaptoethanol) supplemented with cOmplete Protease Inhibitor (Roche) and Phosphatase Inhibitor (Roche). After sonication, the cell extract was centrifuged at 15,000 × g for 30 min at 4°C and the supernatant was isolated. The supernatant was then incubated with GFP-trap Magnetic Agarose (Chromotek), Myc-trap Magnetic Agarose (Chromotek), or Anti-FLAG M2 Magnetic Beads (Sigma) for 2 h at 4°C on a rotating wheel. The beads were washed with high-salt buffer (20 mM HEPES (pH 7.0), 400 mM KCl, 5 mM MgCl₂, 10% glycerol, 0.1% Triton X-100, and 1 mM β-mercaptoethanol) supplemented with cOmplete Protease Inhibitor (Roche) and Phosphatase Inhibitor (Roche). The samples were eluted with SDS loading buffer at 95°C for 5 min.

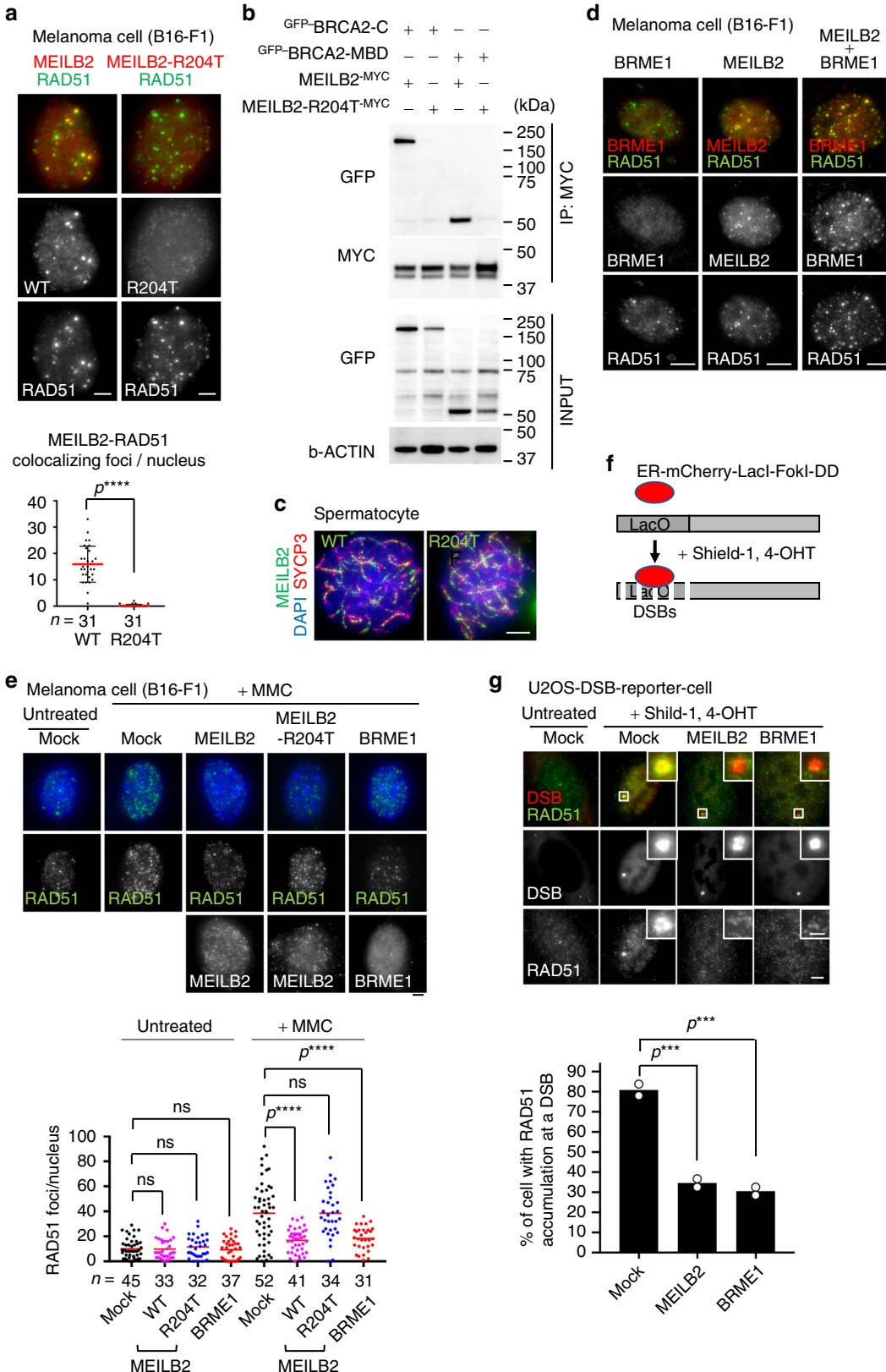

**BRCA2-MEILB2 complex expression and purification.** The pET-Duet vector (Novagen) was used for co-expression of recombinant proteins in *E. coli*. For His-Smt3 tagged proteins in *E. coli* the pSmt3 expression was used, which was obtained from Dr. Christopher D. Lima after signing a material transfer agreement with Cornell University, New York. For co-expression, restriction-based cloning was used to insert MEILB2 (a.a. 87–338) with an N-terminal 10× His-Smt3 tag into the upstream multiple cloning site of the pET-Duet vector, whereas untagged BRCA2

(a.a. 2219–2285) was cloned into the downstream multiple cloning site. For individual protein expression, MEILB2 (a.a. 87–338) and BRCA2 (a.a. 2219–2285) sequences were cloned into the BamH1 and Xho1 sites of the pSmt3 vector. All MEILB2 and BRCA2 constructs were expressed in BL21(DE3) *E. coli* cells for protein purification purposes. For purification of the co-expressed MEILB2 C (a.a. 87–338) and BRCA2-MBD (a.a. 2219–2285) complex, the first step of purification was affinity chromatography using Ni-NTA resin (Qiagen). The Smt3 tag was then

**Fig. 8 MEILB2 and BRME1 impair mitotic HR. a** Mouse B16-F1 cells transfected with MEILB2-MYC (WT) or MEILB2-R204T-MYC (R204T) stained with RAD51 and MYC antibodies. Graph: the number of MYC foci colocalizing with RAD51. *n* shows the analyzed RAD51-positive S to G2 phase cells. Bars: mean values with SD. **b** IPs with the MYC antibody from B16-F1 cells expressing MEILB2-MYC or MEILB2-R204T-MYC with GFP-BRCA2 truncations; C (a.a. 2036–3329) or MBD (a.a. 2117–2339). **c** Immunostaining of zygotene spermatocytes expressing GFP-MEILB2 or GFP-MEILB2-R204T. **d** Mouse B16-F1 cells transfected with FLAG-BRME1 (left), MEILB2-MYC (middle), or both FLAG-BRME1 and MEILB2-MYC (right) and stained with RAD51 and FLAG (for BRME1) or MYC (for MEILB2) antibodies. **e** Mouse B16-F1 cells with or without MMC treatment transfected with MEILB2-MYC (WT or R204T) or FLAG-BRME1 and stained with RAD51 or MYC (for MEILB2) and FLAG (for BRME1) antibodies and DAPI. Graph: the number of RAD51 foci per cell. Red bars: mean values. *n* shows the analyzed cell number pooled from three independent transfections. **f** Schematic of the U2OS-DSB reporter cell line. **g** U2OS-DSB reporter cell line with or without DSB induction transfected with MEILB2-MYC or FLAG-BRME1 and stained with the RAD51 antibody. Graph: the frequency of cells with RAD51 accumulation at the DSB. The analyzed cell numbers are 78, 70, and 68 cells for mock, MEILB2, and BRME1, respectively. The mean values of two independent experiments are shown. All analyses used two-tailed *t* tests. ns: not significant. ***p < 0.001, ****p < 0.0001. Scale bars: 5 μm or 1 μm in the magnified panel. Source data are provided as a Source Data file.

cleaved with Ulp1 protease, and the mixture was run through a cation exchange column (HiTrap SP, GE Healthcare). Size-exclusion chromatography (Superdex 200, GE Healthcare) was used as a final step of purification to obtain a homogenous preparation of the complex as assessed by Coomassie-stained sodium dodecyl sulphate–polyacrylamide gel electrophoresis gels. Individually expressed MEILB2 (a.a. 87–338) and BRCA2 (a.a. 2219–2285) proteins were initially purified using Ni-NTA affinity chromatography. The Smt3 tag was then cleaved with Ulp1 protease and the mixture was resolved using size-exclusion chromatography (Superdex 75 for mBRCA2 a.a. 2219–2285 and Superdex 200 for mMEILB2 a.a. 87–338, GE Healthcare). Each protein was further purified using cation exchange chromatography (HiTrap SP, GE Healthcare). Both size-exclusion chromatography and ion exchange were performed on an AKTA PURE-M protein purification system (GE Healthcare).

**SEC-MALS for the BRCA2-MEILB2 complex.** For molar mass determination of proteins and complexes, 500 μg of protein was loaded onto a Superdex 200 10/300 column (GE Healthcare) at a flow rate of 0.5 ml min$^{-1}$ on an AKTA PURE-L protein purification system connected upstream of a DAWN HELIOS II MALS detector (Wyatt Technology) and an Optilab T-rEX differential refractometer (Wyatt Technology). After exiting the column, a versatile valve diverted the eluates to the MALS detector and the differential refractometer before collecting them using the F9-R fraction collector of the AKTA PURE-L system. ASTRA 6 software (Wyatt Technology) was used to analyze the data. Molecular weights were calculated for protein and protein complex peaks by extrapolation from Zimm plots. Differential refractometry was used to determine the protein concentration, and a dn/dc value of 0.185 ml g$^{-1}$ was used to determine the molecular weight of the protein or complex.

**MEILB2-BRME1 complex expression and purification.** Sequences corresponding to regions of mouse *Meilb2* and *Brme1* were cloned into pMAT11[37] and pET28c+ (Novagen) for expression with an N-terminal TEV-cleavable MBP-tag and non-cleavable His-tag, respectively. Constructs were co-expressed in BL21 (DE3) cells (Novagen) in 2× YT media and induced with 0.5 mM IPTG for 16 h at 25°C. Cell disruption was achieved by sonication in 20 mM Tris pH 8.0 and 500 mM KCl, and cellular debris was removed by centrifugation at 40,000 × *g*. Fusion proteins were purified through consecutive Ni-NTA (Qiagen), amylose (NEB), and HiTrap Q HP (GE Healthcare) ion exchange chromatography. TEV protease was used to remove affinity tags, and cleaved samples were purified through ion exchange chromatography and size-exclusion chromatography (HiLoad 16/600 Superdex 200, GE Healthcare) in 20 mM Tris pH 8.0, 150 mM KCl, and 2 mM DTT. Protein samples were concentrated using Microsep Advance Centrifugal Devices 3000 MWCO centrifugal filter units (PALL) and were stored at −80°C following flash-freezing in liquid nitrogen. Concentrations were determined by UV spectroscopy using a Cary 60 UV spectrophotometer (Agilent) with extinction coefficients and molecular weights calculated by ProtParam (http://web.expasy.org/protparam/) or by measuring the differential refractive index using an Optilab T-rEX differential refractometer (Wyatt Technology).

**Circular dichroism (CD) spectroscopy.** Far UV CD spectroscopy data were collected on a Jasco J-810 spectropolarimeter (Institute for Cell and Molecular Biosciences, Newcastle University). Buffer-subtracted CD spectra were recorded using a 0.2 mm path length cuvette (Hellma) in 10 mM Na$_2$HPO$_4$/NaH$_2$PO$_4$ (pH 7.5) and 250 mM NaF between 260 nm and 185 nm. Data were collected at 20 nm/min with a pitch of 0.2 nm, smoothed with a response time of 4 s, and plotted as mean residue ellipticity ([θ]) (×1000 deg cm$^2$ dmol$^{-1}$ residue$^{-1}$). The CDSSTR algorithm of the Dichroweb server (http://dichroweb.cryst.bbk.ac.uk)[38] was used to estimate the secondary structure content of the analyzed samples. Protein stability was assessed through thermal denaturation in 20 mM Tris (pH 8.0), 150 mM KCl, and 2 mM DTT using a 1 mm path length quartz cuvette (Hellma). Data were recorded at 222 nm between 5°C and 95°C at 1°C per minute every 0.2°C and were converted to mean residue ellipticity ([θ$_{222}$]) and plotted as % unfolded

([θ]$_{222,x}$ − [θ]$_{222,5}$)/([θ]$_{222,95}$ − [θ]$_{222,5}$). Melting temperatures (Tm) were estimated as the points at which samples were 50% unfolded.

**SEC-MALS for the MEILB2-BRME1 complex.** The absolute molar masses of protein samples and complexes were determined by SEC-MALS. Protein samples at >1 mg/ml were loaded onto a Superdex 200 Increase 10/300 GL size-exclusion chromatography column (GE Healthcare) in 20 mM Tris (pH 8.0), 150 mM KCl, and 2 mM DTT at 0.5 ml/min using an ÄKTA Pure (GE Healthcare). The column outlet was fed into a DAWN HELEOS II MALS detector (Wyatt Technology) followed by an Optilab T-rEX differential refractometer (Wyatt Technology). Light scattering and differential refractive index (dRI) data were collected and analyzed using ASTRA 6 software (Wyatt Technology). Molecular weights and estimated errors were calculated across eluted peaks by extrapolation from Zimm plots using a dn/dc value of 0.1850 ml/g. SEC-MALS data are presented with dRI profiles with fitted molecular weights (M$_W$) plotted across elution peaks.

**SEC-SAXS for the MEILB2-BRME1 complex.** SEC-SAXS (Size-exclusion chromatography small angle X-ray scattering) experiments were performed at beamline B21 of the Diamond Light Source synchrotron facility (Oxfordshire, UK). Protein samples at concentrations >10 mg/ml were loaded onto a Superdex 200 Increase 10/300 GL size-exclusion chromatography column (GE Healthcare) in 20 mM Tris (pH 8.0) and 150 mM KCl at 0.5 ml/min using an Agilent 1200 HPLC system. The column outlet was fed into the experimental cell, and SAXS data were recorded at 12.4 keV, detector distance 4.014 m, in 3.0 s frames. Data were subtracted, averaged, and analyzed for Guinier region *Rg* using ScÅtter 3.0 (http://www.bioisis.net), and P(r) distributions were fitted using PRIMUS[39]. The radius of gyration (*Rg*) describes the distribution of atoms within a molecule and is mathematically defined as the root mean square distance from each atom to the center of mass, and thus it indicates the size and shape of a molecule. Guinier analysis estimates the *Rg* from the SAXS scattering curve through determination of the gradient (-*Rg*/3) of a linear portion in the low-Q region of a plot of ln(*I*(*Q*)) against *Q*$^2$, known as the Guinier region. Ab initio modeling was performed using DAMMIF[40]. Thirty independent runs were performed in P1 symmetry and averaged. Molecular surface images were generated using the PyMOL Molecular Graphics System, Version 2.3.2 (Schrödinger, LLC).

**Statistics and reproducibility.** The experiments were not randomized, so no statistical method was used to predetermine sample size, and the investigators were not blinded to allocation during the experiments or to outcome assessment. Each conclusion in the manuscript was based on results that were reproduced in at least two independent experiments. Sample sizes, statistical tests, and *p* values are indicated in the text, figures, and figure legends.

**Reporting summary.** Further information on research design is available in the Nature Research Reporting Summary linked to this article.

## Data availability

Data supporting the findings of this manuscript are available from the corresponding author upon reasonable request. A reporting summary for this Article is available as a Supplementary Information file. The source data underlying Figs. 1a, 3a, b, e, f, 4f, h, i, k, l, 5a–d, 6a–d, 7a, b, 8a, e, g, and Supplementary Figs. 3a, 4e, and 5 are provided as a Source Data file. SEC-SAXS data and models have been deposited in the Small Angle Scattering Biological Data Bank (https://www.sasbdb.com/) under accession codes SASDH99, SASDHA9, SASDHB9, and SASDHC9. All data are available from the authors upon reasonable request.

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

## Acknowledgements

We thank the Karolinska Center for Transgene Technologies, Stockholm, Sweden, for producing the genetically modified mice. We thank Hyunsook Lee (Seoul National University) and Roger A. Greenberg (University of Pennsylvania) for providing materials. We thank the Proteomics Core Facility of the University of Gothenburg for performing the mass spectrometry analysis. We thank Yoshinori Watanabe (Francis Crick Institute) and Yuki Okada (University of Tokyo) for supporting the initial study. We thank Attila Tóth (Technische Universität Dresden), Shintaro Yamada, and Scott Keeney (Memorial Sloan Kettering Cancer Center) for valuable discussions. O.R.D. is a Sir Henry Dale Fellow jointly funded by the Wellcome Trust and Royal Society (Grant number 104158/Z/14/Z). This work was supported by NIH/NIGMS grant R35GM118052 (P.J.W.), the European Research Council StG-801659 (H.S.), the Swedish Research Council 2018-03426 (H.S.), and Cancerfonden 2018/326 (H.S.). Open access funding provided by University of Gothenburg.

## Author contributions

J.Z. performed most of the mouse experiments and analyzed the data; J.Z., K.Z., and M.E. performed the cancer cell analysis; Y.F. contributed to the initial identification of *Brme1*; E.V. analyzed the mass spectrometry data; G.L., M.E., R.G., and P.J.W. analyzed the *Meiob*−/− mice; D.F.P., I.A., and J.N. purified and analyzed the BRCA2-MEILB2 heterocomplex in vitro; M.G. and O.R.D. purified and analyzed the MEILB2-BRME1 heterocomplex in vitro; and H.S. conceptualized and supervised the project and wrote the manuscript.

## Competing interests

The authors declare no competing interests.
