## [Peer Review File · Nature Communications]

Reviewers' comments:

Reviewer #1 (Remarks to the Author):

The Ms reports *in vitro* and *in vivo* studies of the breast cancer susceptibility gene II (BRCA2) binding to its partner protein MEILB2, and further formation of the ternary complex with the associating protein 1 (BRME1). Destabilization of the BRCA2-MEILB2 complex in the absence of BRME1 leads to defects in the repair of DNA double stranded breaks and other DNA damages. The results suggest that the misregulation of the BRCA2 complexes plays a role in cancer development.

The paper is devoted to medically relevant protein complex and utilizes a combination of various approaches including biophysical, biochemical, genetic tools. Overall, the presented results provide new insights in the structural organization and function of the complex and the Ms should be of interest for a broad readership of Nature Communications. Below, I provide comments on the biophysical *in vitro* part of the work.

The authors clearly overestimate the accuracy of the molecular weight determination with SEC-MALS reporting the values up to significant digits after comma.

The structural results are mostly based on SEC-SAXS data revealing low resolution shapes of different constructs. In general, the results appear reasonable and the shapes consistent. As a comment, the authors call Alpha1+2 construct "aggregate". The term "aggregate" in SAXS is usually related to unspecific aggregation. In this case, from the data (a linear Guinier plot in Fig S1f) and from the models presented, Alpha1+2 appears as a specific associate. I would therefore not use the term "aggregate", but e.g. tetramer or associate. Further, an attempt should be made to construct a model of Alpha1+2 tetramer from the shapes of two Alpha1+2 dimers and verify whether this model would fit the experimental data from Alpha1+2 tetramer.

The Ms would have gained a lot from the structural modelling of the ternary complex, however, the conclusions are only made based on the MW values estimated from SEC-MALS. What is the reason for this?

According to the publication guidelines for SAXS data (Trehella et al, Acta Crystallogr D Struct Biol. 2017, 73: 710-728, <https://www.ncbi.nlm.nih.gov/pubmed/28876235>) the data statistics should be summarized in a Table. Further, the SAXS-based models must be deposited in a data bank (SASBDB, <https://www.sasbdb.org>)

Minor comments:

In the abstract, BRME1 is first written in capital letters, then in lowercase italics -- the difference should be explained and Brme1-/- should be defined.

In supplementary Fig. 1d, the ordinate axis is called "Molecular weight (kDa)", should be "Molecular weight (kDa)". On the same Figure, the color of MBP Alpha1+2 (caption)

does not correspond to the color of the elution profile.
There is also "Molecular weight" in the legend of Fig.2c.

The basic concept of "radius of gyration" should be at least introduced in Methods. The same for "Guinier region".

The program DAMMIF used for shape determination is not referenced.

With the color selection in Fig 1j it is difficult to distinguish between the different $p(r)$ functions.

Reviewer #2 (Remarks to the Author):

The BRCA2-MEILB2-BRME1 complex governs meiotic recombination and impairs the mitotic BRCA2-RAD51 axis
Zhang et al. (Shibuya)

Mechanisms that regulate homology-directed repair of meiotic double-strand breaks (DSBs) are poorly understood, particularly in mammals. In a previous study, the authors identified MEILB2, a meiosis specific factor that interacts with BRCA2 and is required for localization of RAD51 and DMC1 strand-exchange proteins. Here the authors build upon this regulatory network. They identify a MEILB2-interactor, BRME1, that stabilizes MEILB2 and show that BRCA2, MEILB2 and BRME1 likely form a complex. A particularly interesting finding is that spermatocytes lacking BRME1 progress to late meiotic prophase stages with unrepaired breaks and experience a boost in RAD51 accumulation that eventually leads to some form of a repair (loss of RPA2 and γ H2AX), however fails to yield normal levels of crossovers. As a result, spermatocytes arrest at metaphase and males are sterile. This work highlights how BRCA2 function is modulated during meiosis and how it may be deregulated in tumorigenic conditions.

Few comments that could improve this study are listed below.

Major points:

1. The authors should list the factors that were enriched in BRME1 IP-MS (Figure 2e), or at least specify whether other known repair factors were present, in addition to BRME1-MEILB2-BRCA2 and SPATA22-MEIOB. For example, was RAD51 present, given that it can be detected in westerns? Was RPA2 present, which would be interesting to know since it may be interacting via SPATA22-MEIOB?
2. There is often the assumption that axis-associated foci in mutant scenarios are comparable to those formed in wild type. In Figure 3d, the conclusion that BRME1-MBD is the minimum region capable of recombination nodule localization requires testing whether the foci formed colocalize with a recombination marker, e.g. RPA2. Although BRME1-MBD forms foci along axes, these may not represent recombination nodules. On a similar note, the data in Figure 3 g,h show that MEILB2 and BRME1 foci are reduced in intensity in mice lacking MEIOB. The authors conclude that SPATA22-MEIOB is not necessary for but facilitates MEILB2-BRME1 recruitment. The weak foci formed may represent mislocalized protein and colocalization with RPA2 is required to support their conclusion. Similarly, the authors conclude that MEILB2-R204T is capable of localizing to meiotic DSBs (Figure 7c); this also needs to be tested by colocalization analyses with RPA2.
3. BRME1 shows MEILB2-dependent localization at mitotic and meiotic DSBs and is unable to directly interact with BRCA2. Thus it is unclear how its overexpression in mitotic cells (that lack MEILB2) leads to reduced RAD51 foci formation upon exogenous damage (Figure 7e,g). Please explain.

4. On page 17 lines 380-384: these statements of the discussion section need to be corrected/clarified. For instance, hotspots are known to be present outside of the PAR (SSDS maps by Brick et al. / Spo11-oligo maps by Lange et al. / RAD51 foci formation by multiple groups).

Additional minor points:

5. BRME1 stabilizes MEILB2 in vitro and in meiosis. Is this also the case in mitotic cells? MEILB2 forms foci that colocalize with RAD51 in cell lines (Figure 7a, 7d). Are these foci brighter when coexpressed with BRME1? Please explain.

6. It is interesting that spermatocytes lacking BRME1 do not elicit a pachytene arrest. Has this been carefully assessed? Do all seminiferous tubules show normal progression until metaphase? If so, this should be mentioned and discussed.

7. It is impossible to see the tubule stage in Figure 6h.

8. The use of 'BRCA2-RAD51 axis' in the title is confusing. Especially given that 'axis' is used to refer to the meiotic chromosome structure.

9. On page 8, the authors conclude that BRCA2 forms a complex that excludes DMC1. Lack of detection in the IP does not necessarily imply lack of interaction. While this is clarified in the discussion, perhaps the conclusion can be softened here too.

10. RAD51 foci counts are incorrectly colored in Figure 4I (zygonema).

11. On page 11, the authors suggest a model where MEILB2-BRME1 'actively' removes SPATA22-MEIOB. The data are consistent with this model but the use of 'active' implies an activity that has not been demonstrated and should be corrected.

We thank the Editor and Reviewers for providing constructive comments on our manuscript. Please find below our point-by-point responses to all the reviewer's comments.

Reviewer #1:

The authors clearly overestimate the accuracy of the molecular weight determination with SEC-MALS reporting the values up to significant digits after comma.

We agree that SEC-MALS data in Fig. 2c have been reported to a higher level of accuracy than is appropriate. We have cut the digits after the decimal point in Fig. 2c and in the related text.

As a comment, the authors call Alpha1+2 construct "aggregate". The term "aggregate" in SAXS is usually related to unspecific aggregation. In this case, from the data (a linear Guinier plot in Fig S1f) and from the models presented, Alpha1+2 appears as a specific associate. I would therefore not use the term "aggregate", but e.g. tetramer or associate.

We have changed the term "aggregate" into "associate" or "octamer" dependent on the context throughout the manuscript and figures.

Further, an attempt should be made to construct a model of Alpha1+2 tetramer from the shapes of two Alpha1+2 dimers and verify whether this model would fit the experimental data from Alpha1+2 tetramer.

The MEILB2 Alpha 1+2 associate is an octamer (four MEILB2 Alpha 1+2 dimers). Examination of the ab initio models confirms that the size and shape of the octamer is consistent with lateral self-association of four dimers (Fig. 1k). To further clarify this point we have added 50 Å scale bars to each inset panel. We had attempted multi-phase ab initio modelling by combining the octamer and dimer datasets, and we were able to obtain good fits, in support of the octamer consisting of four dimers.

The Ms would have gained a lot from the structural modelling of the ternary complex, however, the conclusions are only made based on the MW values estimated from SEC-MALS. What is the reason for this?

We have chosen not to perform any more detailed structural modelling of our protein complexes as we lack the clear structural homologies necessary for modelling, so any models generated would be highly speculative and potentially misleading to readers. Instead, we are attempting protein crystallization of our protein complexes and, if succeeds, will report the detailed structural analyses in the future studies.

According to the publication guidelines for SAXS data (Trehwella et al, Acta Crystallogr D Struct Biol. 2017, 73: 710-728, <https://www.ncbi.nlm.nih.gov/pubmed/28876235>) the data statistics should be summarized in a Table. Further, the SAXS-based models must be deposited in a data bank (SASBDB, <https://www.sasbdb.org>)

We have included a table in Supplementary Fig. 1g that summarizes the main analyses and modelling statistics relating to the SEC-SAXS data as below.

Table 1 Summary of SEC-SAXS data

	MEILB2 $\alpha 1+2$ (18-122) +BRME1	MEILB2 $\alpha 1+2$ (18-122) dimer	MEILB2 $\alpha 1+2$ (18-122) octamer	MEILB2 $\alpha 2$ (51-122) monomer
Guinier analysis				
$I(0)$ (cm ⁻¹)	0.095	0.041	0.022	0.027
R_g (Å)	50	46	57	30
q_{min} (Å ⁻¹)	0.012	0.0091	0.0080	0.018
$P(r)$ analysis				
$I(0)$ (cm ⁻¹)	0.097	0.041	0.022	0.027
R_g (Å)	53	48	59	30
D_{max} (Å)	190	160	200	110
Porod volume (Å ³)	103618	87797	368351	43352
MW from Porod volume (kDa)	61	52	217	26
V_c (Å ²)	569	521	885	212
MW from V_c (kDa)	53	48	112	12
DAMMIF ab initio modelling (30 models)				
Symmetry	P1	P1	P1	P1
NSD mean	0.794	0.898	1.011	0.746
χ^2 (reference model)	1.23	1.31	0.973	1.05

We are further happy to follow the reviewer's suggestion of submitting the data and models to the SASBDB database. We will submit all data relating to this work upon acceptance so we can include the final manuscript title and metadata.

In the abstract, BRME1 is first written in capital letters, then in lowercase italics -- the difference should be explained

According to the rules for gene and protein nomenclature (<https://www.biosciencewriters.com/Guidelines-for-Formatting-Gene-and-Protein-Names.aspx>), the symbols for mouse genes are italicized and the first letter is upper-case (e.g., *Brme1*), while symbols for mouse proteins are not italicized and all letter is upper-case (e.g., BRME1). The symbols for human gene are italicized and the all letter is upper-case (e.g., *BRCA2*), while symbols for human proteins are not italicized and all letter is upper-case (e.g., BRCA2). We are following this nomenclature throughout the manuscript.

For example:

“Here, we identify BRCA2 and MEILB2-associating protein 1 (BRME1)”

In this context, BRME1 is mouse protein and, then, is not italicized and all letter is upper-case.

“In *Brme1* knockout (*Brme1*^{-/-}) mice, the BRCA2-MEILB2 complex is destabilized”

In this context, we are mentioning *Brme1* gene knockout in mice and, therefore, *Brme1* is italicized and the first letter is upper-case.

The above nomenclature is generally accepted and we think we do not need to define the nomenclature in our particular manuscript.

Brme1^{-/-} should be defined.

We have changed the text in the abstract as below.

Changed

“In *Brme1* knockout (*Brme1*^{-/-}) mice, the BRCA2-MEILB2 complex is destabilized”

In supplementary Fig. 1d, the ordinate axis is called "Molecular wright (kDa)", should be "Molecular weight (kDa)". On the same Figure, the color of MBP Alpha1+2 (caption) does not correspond to the color of the elution profile. There is also "Molecular wright" in the legend of Fig.2c.

Thank you for pointing it out. We have corrected the color and typos.

The basic concept of "radius of gyration" should be at least introduced in Methods. The same for "Guinier region".

We have added the below sentences in the methods section.

Added

“The radius of gyration (*R_g*) describes the distribution of atoms within a molecule, and is mathematically defined as the root mean square distance from each atom to the centre of mass, so indicates the size and shape of a molecule. Guinier analysis estimates the *R_g* from the SAXS scattering curve through determination of the gradient ($-R_g/3$) of a linear portion in the low-*Q* region of a plot of $\ln(I(Q))$ against Q^2 , known as the Guinier region.”

The program DAMMIF used for shape determination is not referenced.

We have added the below reference in the method section.

Added

Reference 40 “Franke, D. & Svergun, D.I. DAMMIF, a program for rapid ab-initio shape determination in small-angle scattering. *J Appl Crystallogr* **42**, 342-346 (2009).”

With the color selection in Fig 1j it is difficult to distinguish between the different *p*(*r*) functions.

We have changed the color as below.

Reviewer #2:

1. The authors should list the factors that were enriched in BRME1 IP-MS (Figure 2e), or at least specify whether other known repair factors were present, in addition to BRME1-MEILB2-BRCA2 and SPATA22-MEIOB. For example, was RAD51 present, given that it can be detected in westerns? Was RPA2 present, which would be interesting to know since it may be interacting via SPATA22-MEIOB?

We are still analyzing some of the uncharacterized proteins identified by the IP-MS/MS and want to publish the results in the future, so that we would like to keep the whole gene list confidential at this point.

However, as requested, all known DNA-damage repair factors identified by our IP-MS/MS analysis has been already highlighted in Fig. 2e. That means we have identified only BRME1, BRCA2, MEILB2, SPATA22, and MEIOB but not the other known repair factors including RAD51 or RPA2 in our IP-MS/MS analysis. The Western-blot is much more sensitive than IP-MS/MS, that is likely the reason why RAD51 was detected by Western-blot but not by the IP-MS/MS. As requested, we have specified this point in Figure legend in Fig. 2e as below.

Added

“Genes involved in meiotic HR regulation are indicated. Note the other DNA repair factors, such as RAD51, were not detected by the mass spectrometry analysis, likely due to the limited sensitivity.”

2. There is often the assumption that axis-associated foci in mutant scenarios are comparable to those formed in wild type. In Figure 3d, the conclusion that BRME1-MBD is the minimum region capable of recombination nodule localization requires testing whether the foci formed colocalize with a recombination marker, e.g. RPA2. Although BRME1-MBD forms foci along axes, these may not represent recombination nodules.

We have added a data showing the triple staining of GFP-BRME1-MBD, RPA2, and SYCP3 in Supplementary Fig. 3b. We can see that GFP-BRME1-MBD and RPA2 signals are largely colocalizing indicating that GFP-BRME1-MBD localizes to the recombination nodules.

On a similar note, the data in Figure 3 g,h show that MEILB2 and BRME1 foci are reduced in intensity in mice lacking MEIOB. The authors conclude that SPATA22-MEIOB is not necessary for but facilitates MEILB2-BRME1 recruitment. The weak foci formed may represent mislocalized protein and colocalization with RPA2 is required to support their conclusion.

We have added a data showing the triple staining of MEILB2/BRME1, RPA2, and SYCP3 in *Meiob* knockout mice in Supplementary Fig. 3e. We can see that the faint MEILB2/BRME1 signals still colocalizes with RPA2, suggesting that MEILB2/BRME1 localizes to the recombination nodules even in *Meiob* knockout mice.

Similarly, the authors conclude that MEILB2-R204T is capable of localizing to meiotic DSBs (Figure 7c); this also needs to be tested by colocalization analyses with RPA2.

We have added a data showing the triple staining of GFP-MEILB2-R204T, RPA2, and SYCP3 in Supplementary Fig. 7b. We can see of GFP-MEILB2-R204T and RPA2 signals are largely colocalizing indicating that GFP-MEILB2-R204T localizes to the recombination nodules.

3. BRME1 shows MEILB2-dependent localization at mitotic and meiotic DSBs and is unable to directly interact with BRCA2. Thus it is unclear how its overexpression in mitotic cells (that lack MEILB2) leads to reduced RAD51 foci formation upon exogenous damage (Figure 7e,g). Please explain.

This is an interesting open question newly arose in this study. We are thinking that some somatic HR protein may bind to BRME1 directly and be sequestered from its functional complex, leading to the abolition of HR activation. To prove this hypothesis, we are now trying to identify BRME1 interacting protein in cancer cells by the mass spectrometry followed by BRME1 overexpression and IP. If we succeed to address the mechanisms, we will report it in the future studies. For readers who may have the same question, I have added a sentence explaining this speculation in discussion section as below.

*Added*

*"BRME1 may also exert its inhibitory function by binding to some mitotic DDR proteins. It will be a challenge for future studies to discover the responsible BRME1 partner proteins."*

4. On page 17 lines 380-384: these statements of the discussion section need to be corrected/clarified. For instance, hotspots are known to be present outside of the PAR (SSDS maps by Brick et al. / Spo11-oligo maps by Lange et al. / RAD51 foci formation by multiple groups).

Thank you for pointing it out. For accuracy, we have deleted the sentence discussing the DSB hotspot on sex chromosomes as below.

*Deleted*

*"The DSB hotspots on sex chromosomes are located only at the PAR in WT, and it would be interesting to investigate whether cryptic DSB hotspots can be generated at other than the PAR in Brme1−/− spermatocytes. Alternatively, the position of DSB hotspots might be unchanged, but DSBs in non-hotspot regions can be used for strand invasion due to the defective homology search."*

5. BRME1 stabilizes MEILB2 in vitro and in meiosis. Is this also the case in mitotic cells? MEILB2 forms foci that colocalize with RAD51 in cell lines (Figure 7a, 7d). Are these foci brighter when coexpressed with BRME1? Please explain.

We are expressing the MEILB2 and BRME1 in mitotic cells by transient transfection and the protein expression level (plasmid copy number) differs between individual cells, making it difficult to interpret the difference in foci brightness between individual transfected cells. However, there is some evidence: if we look at the INPUT of Fig. 2d, MEILB2-MYC bands are somehow stronger when co-expressed with FLAG-BRME1 (lane 4,7 vs 3,6,9), suggesting that the co-expression of BRME1 may somewhat stabilize MEILB2 also in mitotic cells. However, again, this is a transient transfection experiment and we cannot accurately control the copy number of plasmids incorporated into the cells and we would like to avoid the hasty interpretation in the current manuscript.

6. It is interesting that spermatocytes lacking BRME1 do not elicit a pachytene arrest. Has this been carefully assessed? Do all seminiferous tubules show normal progression until metaphase? If so, this should be mentioned and discussed.

As you pointed out, apoptosis happened in two different stages, either in pachytene (stage V-VI) or metaphase I (stage XII). We showed the first population (cell death in pachytene stage) in Figure 4f, while we showed the other population (cell death in metaphase I) in Figure 6h. To clarify this point, we have changed the representative picture of Figure 4f, where we can see the epithelial cycle of seminiferous tubules more easily by the DAPI signals.

In the new picture (also see the below DAPI channel alone), now we can clearly see the type-B spermatogonia (B) at the periphery and pachytene spermatocytes (P) at the second layer of the tubules. TUNEL signal is seen in some, if not all, of the pachytene cells.

We have also changed the representative pictures in Figure 6h, where we can see the cell death in metaphase I more clearly (see the next section). Further, we have added a text in page 14 in order to make it clear that cell death happens in two different stages.

Added

*“TUNEL assay showed significant cell death in metaphase I spermatocytes at stage XII seminiferous tubules (Fig. 6h). Thus, there are two rounds of cell death in *Brme1*^{-/-} testes; either in pachytene spermatocytes at stage V-VI (Fig 4f) or in metaphase I at stage XII (Fig. 6h). Together these results lead to the conclusion that BRME1 is indispensable for the DSB repair, homologous synapsis, and crossover formation, which are needed for progression past metaphase I (Fig. 6i).”*

Finally, we have also described the epithelial cycle of the seminiferous tubules in figure legend in Fig. 4f and Fig. 6i.

7. It is impossible to see the tubule stage in Figure 6h.

We have changed the representative pictures in Fig. 6h as below.

In the new picture, now we can clearly see the condensed chromosomes in metaphase I spermatocytes suggesting that this tubule is at stage XII (please also find the below picture with DAPI channel alone, where I marked metaphase I cells with asterisks).

To further help the readers, we have added a magnified picture with DAPI channel in Fig. 6h, where we can see TUNEL-positive metaphase I cells with misaligned univalent chromosomes (shown with arrowheads).

8. The use of ‘BRCA2-RAD51 axis’ in the title is confusing. Especially given that ‘axis’ is used to refer to the meiotic chromosome structure.

We have changed the title as below.

Before

“The BRCA2-MEILB2-BRME1 complex governs meiotic recombination and impairs the mitotic BRCA2-RAD51 axis”

After

“The BRCA2-MEILB2-BRME1 complex governs meiotic recombination and impairs the mitotic BRCA2-RAD51 function in cancer cells”

9. On page 8, the authors conclude that BRCA2 forms a complex that excludes DMC1. Lack of detection in the IP does not necessarily imply lack of interaction. While this is clarified in the discussion, perhaps the conclusion can be softened here too.

We have changed the text as below.

Before

“our in vivo data question the physiological significance of the BRCA2-DMC1 interaction.”

After

“our in vivo data suggested that the physiological interaction between BRCA2 and DMC1 is weak or transient.”

10. RAD51 foci counts are incorrectly colored in Figure 4l (zygonema).

Thank you for pointing it out. We have corrected the color as below.

11. On page 11, the authors suggest a model where MEILB2-BRME1 ‘actively’ removes SPATA22-MEIOB. The data are consistent with this model but the use of ‘active’ implies an activity that has not been demonstrated and should be corrected.

We agree with your point. We have changed the sentences as below. Further we have deleted the term “removal” in Fig. 4m in order to make this point clear.

Before

“We previously reported that the signal intensity of SPATA22 foci is significantly stronger in Meilb2^{-/-} compared to WT, suggesting that MEILB2 actively removes SPATA22-MEIOB from ssDNA, likely for the subsequent recruitment of recombinases. Consistent with this, the SPATA22 foci were also stronger in Brme1^{-/-} mice compared to WT, while the degree of increase was less than Meilb2^{-/-} (Supplementary Fig. 4e). Together, these data suggest a model in which MEILB2-BRME1 is initially clamped on ssDNA by SPATA22-MEIOB but later actively removes SPATA22-MEIOB from ssDNA for the loading of recombinases (Fig. 4m).”

After

“We previously reported that the signal intensity of SPATA22 foci is significantly stronger in Meilb2^{-/-} compared to WT. Consistent with this, the SPATA22 foci were stronger also in Brme1^{-/-} mice compared to WT, while the degree of increase was less than Meilb2^{-/-} (Supplementary Fig. 4e). Together, these data suggest a model in which MEILB2-BRME1 is initially clamped on ssDNA by SPATA22-MEIOB and, then, inhibits the excess loading of SPATA22-MEIOB likely by facilitating the loading of recombinases (Fig. 4m).”

REVIEWERS' COMMENTS:

Reviewer #1 (Remarks to the Author):

The authors took into account most of the comments in the revised version of the Ms. Two remarks are still left:

- in Table 1 presenting the SEC-SAXS data, no error bars are given for the results -- they should be added whenever possible. Further, presenting Porod volumes with 5-6 significant digits is unphysical.

- submission of models to SASBDB, similar rules as for PDB hold: the authors are expected to submit the data/models before/during the paper submission. The items are kept on hold before publication and the SASBDB ID, similar to PDB ID assigned upon deposition should be included in the manuscript.

Reviewer #2 (Remarks to the Author):

The authors have addressed my comments satisfactorily. I have no additional concerns.

Response to reviewer #1

- in Table 1 presenting the SEC-SAXS data, no error bars are given for the results -- they should be added whenever possible. Further, presenting Porod volumes with 5-6 significant digits is unphysical.

We have added errors to the relevant statistics in Supplementary Fig. 1g.

We have rounded the Porod volumes to three significant figures.

- submission of models to SASBDB, similar rules as for PDB hold: the authors are expected to submit the data/models before/during the paper submission. The items are kept on hold before publication and the SASBDB ID, similar to PDB ID assigned upon deposition should be included in the manuscript.

We have just deposited the data on SASBDB (please find the below screenshot). It will take some (uncertain) time until we get the accession ID of our deposit data, so that we are not including the ID of the deposited data in our manuscript.

[REDACTED]